# Adalina: Adaptive Linear Approximation for the Shapley Value and Beyond

**Weida Li** [1]   **Yaoliang Yu** [2] [3]   **Bryan Kian Hsiang Low** [1]

## Abstract

The Shapley value, and its broader family of semi-values, has received much attention in various attribution problems. A fundamental and long-standing challenge is their efficient approximation, since exact computation generally requires an exponential number of utility queries in the number of players $n$. To meet the challenges of large-scale applications, we explore the limits of efficiently approximating semi-values under a $\Theta(n)$ space constraint. Building upon a vector concentration inequality, we establish a theoretical framework that enables sharper query complexities for existing unbiased randomized algorithms. Within this framework, we systematically develop a linear-space algorithm that requires $O(\frac{n}{\epsilon^2} \log \frac{1}{\delta})$ utility queries to ensure $P(\|\hat{\phi} - \phi\|_2 \geq \epsilon) \leq \delta$ for all commonly used semi-values. In particular, our framework naturally bridges OFA, unbiased kernelSHAP, SHAP-IQ and the regression-adjusted approach, and definitively characterizes when paired sampling is beneficial. Moreover, our algorithm allows explicit minimization of the mean squared error $\mathbb{E}[\|\hat{\phi} - \phi\|_2^2]$ for each specific utility function. Accordingly, we introduce the first adaptive, linear-time, linear-space randomized algorithm, Adalina, that theoretically achieves improved mean squared error. All of our theoretical findings are experimentally validated. Our code is available at https://github.com/watml/adalina.

## 1. Introduction

In recent years, the Shapley value and its broader family of semi-values have found various potential applications in machine learning (Rozemberczki et al., 2022; Cohen-Wang

[1]Department of Computer Science, National University of Singapore, Republic of Singapore [2]School of Computer Science, University of Waterloo, Canada [3]Vector Institute, Canada. Correspondence to: Weida Li <vidaslee@gmail.com>.

*Proceedings of the 43rd International Conference on Machine Learning*, Seoul, South Korea. PMLR 306, 2026. Copyright 2026 by the author(s).

et al., 2024; Deng et al., 2025). The popularity of the Shapley value mainly comes from its uniqueness in satisfying a certain set of axioms (Shapley, 1953). In certain machine learning applications, the axiom of efficiency is unarguably unnecessary (Kwon & Zou, 2022a;b), and the removal of it leads to a broader family of semi-values (Dubey et al., 1981). Specially, each semi-value $\phi(U) \in \mathbb{R}^n$ can be expressed as, for every $i \in [n] := \{1, 2, \ldots, n\}$,

$$\phi_i(U) := \sum_{S \subseteq [n] \setminus \{i\}} p_{|S|+1}[U(S \cup \{i\}) - U(S)]$$
$$\text{with } p_s = \int_0^1 t^{s-1}(1-t)^{n-s} \mathrm{d}\mu(t), \tag{1}$$

where $\mu$ is any Borel probability measure on the closed interval $[0, 1]$. For the Shapley value, $\mu$ corresponds to the uniform distribution, resulting in $p_s = \frac{1}{n}\binom{n-1}{s-1}^{-1}$. Here, $U: 2^{[n]} \to \mathbb{R}$ is the so-called utility function that depends on the context. For example, in attributing the model performance to each data point, $U(S)$ is usually defined as the performance of models trained on $S$ (Ghorbani & Zou, 2019; Ilyas et al., 2022). In explaining the contribution of each feature to a specific model prediction, $U(S)$ is usually defined as the expected prediction where features not in $S$ are treated as missing (Lundberg & Lee, 2017; Lundberg et al., 2020). From Eq. (1), it is clear that computing $\phi$ exactly in general requires an exponential number of utility queries of $U$, which constitutes the major hurdle that limits the applicability of semi-values.

Therefore, considerable effort has been devoted to designing efficient randomized algorithms for approximation. (Jia et al., 2019; Covert & Lee, 2021; Zhang et al., 2023; Wang & Jia, 2023; Li & Yu, 2023; Kolpaczki et al., 2024; Li & Yu, 2024a; Fumagalli et al., 2024; Li & Yu, 2024b; Musco & Witter, 2025; Chen et al., 2025; Witter et al., 2025). The existing randomized algorithms can be divided into two categories. The first category improves the approximation quality by minimizing the mean squared error (MSE)

$$\mathbb{E}[\|\hat{\phi} - \phi\|_2^2],$$

where $\hat{\phi}$ denotes an estimate of $\phi$ produced by a randomized algorithm. All stratified algorithms (Castro et al., 2017; Zhang et al., 2023; Wu et al., 2023) follow this pattern. The

second category tries to provide sharp query complexities, defined as the minimum number of queries to ensure

$$\mathbb{P}(\|\hat{\phi} - \phi\|_2 \geq \epsilon) \leq \delta$$

for every utility function that satisfies $\|U\|_\infty \leq C$, where $C$ is constant as $n \to \infty$ (Wang & Jia, 2023). Very often, the query complexity is determined by corner-case utility functions satisfying $U(S) \in \{C, -C\}$ for every $S$. These two objectives are connected via Chebyshev's inequality,

$$P(\|\hat{\phi} - \phi\|_2 \geq \epsilon) \leq \frac{\mathbb{E}[\|\hat{\phi} - \phi\|_2^2]}{\epsilon^2}.$$

Suppose $\hat{\phi} = \frac{1}{T}\sum_{t=1}^T \hat{\phi}_t$ where $\{\hat{\phi}_t\}_{t=1}^T$ are i.i.d. unbiased estimate of $\phi$. By requiring $\frac{\mathbb{E}[\|\hat{\phi} - \phi\|_2^2]}{\epsilon^2} \leq \delta$, the query complexity is given by $T \geq \frac{\mathbb{E}[\|\hat{\phi}_t - \phi\|_2^2]}{\epsilon^2 \delta}$. To our knowledge, this is the only known regime where MSE and query complexity can be simultaneously optimized.

To obtain a $\log\frac{1}{\delta}$ dependence rather than $\frac{1}{\delta}$ in the query complexity, Hoeffding-type concentration techniques are typically applied, which in turn rely on independence among $\{\hat{\phi}_t\}_{t=1}^T$.[1] In this regime, we observe that minimizing the MSE does not translate into improved query complexity, as it often introduces dependence among $\{\hat{\phi}_t\}_{t=1}^T$. Surprisingly, to our knowledge, approximation algorithms designed by minimizing the MSE are not even theoretically equipped with clearly improved MSE, the difficulty of which may stem from the introduced convoluted dependence. Moreover, these randomized algorithms come at the expense of using $\Theta(n^2)$ space instead. As such, it remains an open question:

*Is it possible to provably minimize the MSE while maintaining a $\Theta(n)$ space constraint?*

In the last two years, the query complexities for approximating semi-values have seen significant advances. Recently, Li & Yu (2024b) introduced the one-for-all (OFA) algorithm for approximating all semi-values and proved that it achieves a query complexity of $O(\frac{n}{\epsilon^2}\log\frac{n}{\delta})$ for all commonly used semi-values, such as Beta Shapley values (Kwon & Zou, 2022a), which include the Shapley value, and weighted Banzhaf values (Li & Yu, 2023), which include the Banzhaf value (Banzhaf III, 1965). Then, Chen et al. (2025) established a provable framework for all the kernelSHAP variants (Lundberg & Lee, 2017; Covert & Lee, 2021; Musco & Witter, 2025) and demonstrated that, by modifying the sampling distribution, the query complexity of unbiased kernelSHAP in approximating the Shapley

value becomes $O(\frac{n}{\epsilon^2 \delta})$. In particular, OFA consumes $\Theta(n^2)$ space, while the other algorithm uses $\Theta(n)$ space. Then, a natural question arises:

*Can semi-values be approximated with a query complexity of $O(\frac{n}{\epsilon^2}\log\frac{1}{\delta})$ using only $\Theta(n)$ space?*

To meet the challenges of large-scale applications (Cohen-Wang et al., 2024; He et al., 2024; Tian et al., 2026), we limit ourselves to a $\Theta(n)$ space constraint. In this work, we will give an affirmative answer to both questions.

**Our theoretical contributions.** As will be demonstrated later, the modified unbiased kernelSHAP turns out to be the linear-space version of OFA, and the clue is evident, as both share the same sampling distribution. Therefore, the comparison between $O(\frac{n}{\epsilon^2}\log\frac{n}{\delta})$ and $O(\frac{n}{\epsilon^2 \delta})$ suggests that the $\log n$ factor may arise from the limitations of the perspective used. Specifically, this $\log n$ factor comes from the union bound:

$$\mathbb{P}(\|\hat{\phi} - \phi\|_2 \geq \epsilon) \leq \mathbb{P}(\bigcup_{i \in [n]} |\hat{\phi}_i - \phi_i| \geq \frac{\epsilon}{\sqrt{n}})$$
$$\leq n \cdot \mathbb{P}(|\hat{\phi}_1 - \phi_1| \geq \frac{\epsilon}{\sqrt{n}}),$$

which leads to bounding each individual estimate $\hat{\phi}_i$ as a first step. This observation motivates us to consider bounding $\hat{\phi}$ as a whole, where a vector concentration inequality comes into play. Indeed, this perspective enables sharper query complexities for existing unbiased randomized algorithms. For example, the query complexity of the modified unbiased kernelSHAP will be improved to $O(\frac{n}{\epsilon^2}\log\frac{1}{\delta})$.

Building upon this holistic perspective, we will establish a framework for approximating semi-values. As will be shown later, our framework naturally bridges several recent approaches (Li & Yu, 2024b; Fumagalli et al., 2024; Witter et al., 2025; Chen et al., 2025). Not only does our framework provide sharper query complexities for these approaches, but it also offers that:

- For the use of paired sampling (Covert & Lee, 2021), it is theoretically established in Theorem 3.2 that the MSE is improved if $\mathbb{E}[U(\mathbf{S}) \cdot U([n] \setminus \mathbf{S})] > 0$;

- For approximating the Shapley value, the sampling distribution used by OFA and the modified unbiased kernelSHAP is the unique optimal solution that minimizes the query complexity;

- For approximating semi-values, the sampling distribution used by SHAP-IQ does not minimize the query complexity, in contrast to the one used by Witter et al. (2025) and OFA.

---

[1]To our knowledge, this independence assumption may instead be relaxed using the concept of exchangeable pairs (Chatterjee, 2007; Mackey et al., 2014), but we did not notice any additional advantage in our context.

We note that, except for OFA, these approaches only heuristically select the sampling distribution, without demonstrating whether their choices correspond to the unique optimal solution in terms of query complexity.

For the randomized algorithm established in our framework, the query complexity and the MSE are fully decoupled: once the sampling distribution is optimized for query complexity, the MSE can be further optimized solely through $U$. Specifically, to ensure $\mathbb{P}(\|\hat{\phi} - \phi\|_2 \geq \epsilon) \leq \delta$, it requires

$$\frac{4nD^*C^2}{\epsilon^2} \log \frac{2}{\delta} \quad \text{utility queries,}$$

$$\text{and} \quad \mathbb{E}[\|\hat{\phi} - \phi\|_2^2] = \frac{nD^*\mathbb{E}[U(S)^2]}{T} + \text{constant.}$$

In particular, $D^* \in O(1)$ for Beta Shapley values and weighted Banzhaf values. Consequently, minimizing the MSE reduces to minimizing $\mathbb{E}[U(S)^2]$, which is upper bounded by $\|U\|_\infty^2$.

**Our algorithmic contributions.** Another appealing property is that the distribution under $\mathbb{E}[U(S)^2]$ is the same as the one used to approximate $\phi$. It implies that, while approximating $\phi$, we can simultaneously solve the problem

$$\underset{V \in \mathcal{V}}{\text{minimize}} \; \mathbb{E}[(U(\mathbf{S}) - V(\mathbf{S}))^2],$$

where $V \in \mathcal{V}$ satisfies $\phi(V) = \mathbf{0}$. This forms the foundation for the design of our adaptive, linear-time, linear-space randomized algorithm, namely Adalina in Algorithms 1 and 2, which automatically minimizes the MSE for each specific utility function $U$. In particular, we theoretically prove that Adalina improves the MSE while maintaining $O(\frac{n}{\epsilon^2} \log \frac{1}{\delta})$ query complexity, as shown in Theorem 4.1. Our theoretical findings align well with empirical results, and our adaptive randomized algorithm consistently performs well across different utility functions and semi-values.

## 2. Vector Concentration Inequality and Sharper Query Complexities

In this section, we recall a vector concentration inequality and demonstrate its usefulness in providing sharper query complexities for unbiased estimates. For a biased estimate $\hat{\phi}_{\text{biased}}$, such as OFA, establishing its query complexity typically involves constructing an unbiased counterpart $\hat{\phi}_{\text{unbiased}}$, after which the analysis (implicitly) bounds $\|\hat{\phi}_{\text{biased}} - \hat{\phi}_{\text{unbiased}}\|_2$ as an additional term. We note that all existing query complexity analyses for biased estimates proceed in this manner (Wang & Jia, 2023; Li & Yu, 2023; 2024a;b), except for least-square-type estimates (Chen et al., 2025). As a result, the established query complexities for biased estimates are worse than those of their unbiased counterparts. Empirically, however, biased estimates can

perform significantly better than its unbiased counterparts, a phenomenon that so far lacked theoretical explanations.

The following vector concentration inequality is based on Yurinsky (1995, Theorem 3.3.4); see Appendix A for a self-contained proof. The significance of this result is that the dimension of the vectors does not appear at all, which is not the case had we applied the union bound to each coordinate (as in many previous works) or the matrix concentration bound (e.g., Tropp et al., 2015, Theorem 6.1.1).

**Theorem 2.1** (Vector concentration inequality). *Suppose* $\{\mathbf{X}_i\}_{i=1}^M$ *are i.i.d. zero-mean random vectors such that* $\mathbb{E}[\|\mathbf{X}_i\|_2^2] \leq \sigma^2$ *and* $\|\mathbf{X}_i\|_2 \leq C$ *almost surely. Then, for every* $0 < \epsilon \leq \frac{3\sigma^2}{C}$, *there is*

$$\mathbb{P}\left(\left\|\frac{1}{M}\sum_{i=1}^M \mathbf{X}_i\right\|_2 \geq \epsilon\right) \leq 2\exp\left(-\frac{M\epsilon^2}{4\sigma^2}\right).$$

As will be shown below, $\frac{3\sigma^2}{C} \to \infty$ as $n \to \infty$ in our context; hence, this constraint can be ignored when deriving query complexities.

Next, we demonstrate how Theorem 2.1 enables sharper query complexities for unbiased estimates. Take AME (average marginal effect, Lin et al., 2022) as an example, whose established query complexity is $O(\frac{n}{\epsilon^2} \log \frac{n}{\delta})$ for its unbiased variant approximating weighted Banzhaf values (Li & Yu, 2024b, Proposition 6). Note that the unbiased variant is also the unbiased counterpart for analyzing the maximum sample reuse (MSR) approach in Li & Yu (2023); Wang & Jia (2023).

The unbiased AME estimator for $w$-weighted Banzhaf value works as follows: First, we randomly sample a subset $S \subseteq [n]$ by including each player independently with probability $w$ (recall $0 < w < 1$). Then, the random vector $\mathbf{X} \in \mathbb{R}^n$ defined as

$$\mathbf{X}_i = \frac{1}{w} \cdot U(\mathbf{S}) \cdot [\![i \in \mathbf{S}]\!] - \frac{1}{1-w} \cdot U(\mathbf{S}) \cdot [\![i \notin \mathbf{S}]\!]$$

is an unbiased estimate of the $w$-weighted Banzhaf value $\phi$. Indeed, let $s = |S|$ be the cardinality, we verify

$$\mathbb{E}[\mathbf{X}_i] = \sum_{S \subseteq [n]} U(S) \cdot w^s(1-w)^{n-s} \cdot \left(\frac{[\![i \in S]\!]}{w} - \frac{[\![i \notin S]\!]}{1-w}\right)$$

$$= \sum_{S \not\ni i} w^s(1-w)^{n-s-1}[U(S \cup i) - U(S)] =: \phi.$$

To reduce the variance, we average over $T$ i.i.d. copies $\{\mathbf{X}_t\}_{t=1}^T$ and obtain $\hat{\phi}^{\text{AME}} := \frac{1}{T}\sum_{t=1}^T \mathbf{X}_t$. Since $\|\phi\|_2 = \|\mathbb{E}[\mathbf{X}_t]\|_2 \leq \mathbb{E}[\|\mathbf{X}_t\|_2]$ and $\|\mathbf{X}_t\|_2^2 \leq c^2 := \frac{nC^2}{w^2 \wedge (1-w)^2}$, (assuming that $U(S) \leq C$), we have

$$\|\mathbf{X}_t - \phi\|_2 \leq 2c \quad \text{and} \quad \mathbb{E}[\|\mathbf{X}_t - \phi\|_2^2] \leq c^2.$$

Applying Theorem 2.1, for every $\epsilon \leq \frac{3}{2}c$, we have

$$\mathbb{P}(\|\hat{\phi}^{\mathrm{AME}} - \phi\|_2 \geq \epsilon) \leq 2\exp\left(-\frac{T\epsilon^2}{4c^2}\right) =: \delta,$$

leading to $T \geq \frac{4nC^2}{\epsilon^2[w^2 \wedge (1-w)^2]}\log\frac{2}{\delta}$. Therefore, we have improved the query complexity of unbiased AME from $O(\frac{n}{\epsilon^2}\log\frac{n}{\delta})$ to $O(\frac{n}{\epsilon^2}\log\frac{1}{\delta})$, amounting to removing a log factor but more importantly revealing that unbiased AME is already a linear-time, linear-space algorithm for approximating weighted Banzhaf values.

Below, we will repeatedly apply Theorem 2.1 to derive query complexities in the same way as illustrated above.

## 3. Our Framework for Approximating Semi-Values

We begin by rewriting the formula of semi-values in Eq. (1):

$$\phi = \varphi + (p_n u_{[n]} - p_1 u_{\emptyset})\mathbf{1}_n \text{ with } \varphi := \sum_{\emptyset \subsetneq S \subsetneq [n]} u_S \mathbf{w}_S,$$

where $u_S := U(S)$ and

$$(\mathbf{w}_S)_i := p_s [\![i \in S]\!] - p_{s+1}[\![i \notin S]\!].$$

Throughout, for a set $S$ we use the corresponding lowercase $s$ to denote its cardinality and the Iverson bracket $[\![A]\!]$ equals 1 if $A$ holds and 0 otherwise.

Since calculating $p_n u_{[n]} - p_1 u_{\emptyset}$ costs only 2 utility evaluations, we will focus on the approximation of $\varphi$.

To sample a (nonempty and proper) subset $S \subseteq [n]$, we first sample its size $s \in [n-1]$ according to a probability vector $\mathbf{q} \in \mathbb{R}^{n-1}$. Then, we sample $S$ uniformly from all subsets with size $s$. Given a sequence of such sampled subsets $\{S_t\}_{t=1}^T$, we form an *unbiased* estimate of $\phi$:

$$\hat{\phi} = \frac{1}{T}\sum_{t=1}^T \frac{u_{S_t}}{q_{s_t}\binom{n}{s_t}^{-1}}\mathbf{w}_{S_t} + (p_n u_{[n]} - p_1 u_{\emptyset})\mathbf{1}_n. \quad (2)$$

Let $m_s = \binom{n-1}{s-1}p_s$ so that $\sum_{s=1}^n m_s = 1$ according to Eq. (1). Then, we have

$$(\mathbf{z}_S)_i := \frac{\binom{n}{s}}{q_s}(\mathbf{w}_S)_i = \frac{n}{q_s}\left(\frac{m_s}{s}[\![i \in S]\!] - \frac{m_{s+1}}{n-s}[\![i \notin S]\!]\right).$$

Consequently,

$$\hat{\phi} = \frac{1}{T}\sum_{t=1}^T u_{S_t}\mathbf{z}_{S_t} + (m_n u_{[n]} - m_1 u_{\emptyset})\mathbf{1}_n, \quad (3)$$

whence Theorem 2.1 applies. To analyze its query complex-

ity, we note that

$$\mathbb{E}[\|u_{\mathbf{S}}\mathbf{z}_{\mathbf{S}}\|_2^2] = \sum_{\emptyset \subsetneq S \subsetneq [n]} q_s \binom{n}{s}^{-1}\frac{n^2}{q_s^2}\left(\frac{m_s^2}{s} + \frac{m_{s+1}^2}{n-s}\right)u_S^2$$

$$= n \cdot D(\mathbf{q}) \cdot \mathbb{E}_{\tilde{q}}[u_{\mathbf{S}}^2] \leq n \cdot D(\mathbf{q}) \cdot C^2$$

where

$$\tilde{q}_s \propto \frac{n}{q_s}\left(\frac{m_s^2}{s} + \frac{m_{s+1}^2}{n-s}\right)$$

and $D(\mathbf{q}) := \sum_{s=1}^{n-1}\frac{n}{q_s}\left(\frac{m_s^2}{s} + \frac{m_{s+1}^2}{n-s}\right)$

Throughout we assume $\|\mathbf{u}\|_\infty \leq C$ for some constant $C$ (that does not depend on $n$).

Although developed differently, the term $D(\mathbf{q})$ also appeared in the OFA (one-for-all) framework, where it is employed to optimize the associated query complexity (Li & Yu, 2024b, Theorem 1). This is not a coincidence, as our framework can be derived from OFA by reducing its space complexity to $\Theta(n)$. We refer the reader to Appendix B for more details.

According to Theorem 2.1, when $\epsilon$ is sufficiently small, the unbiased estimator in Eq. (3) requires at most

$$\frac{4nD(\mathbf{q})\mathbb{E}_{\tilde{q}}[u_{\mathbf{S}}^2]}{\epsilon^2}\log\frac{2}{\delta} \leq \frac{4nC^2 D(\mathbf{q})}{\epsilon^2}\log\frac{2}{\delta}$$

utility queries to ensure $\mathbb{P}(\|\hat{\phi} - \phi\|_2 \geq \epsilon) \leq \delta$. Clearly, $D(\mathbf{q})$ is the dominant factor governing the query complexity, whereas $\mathbb{E}_{\tilde{q}}[u_{\mathbf{S}}^2]$ determines the MSE. Remarkably, our framework achieves linear query complexity if $D(\mathbf{q}) \in O(1)$. Using the Cauchy-Schwartz inequality,

$$D(\mathbf{q}) \geq \left(\sum_{s=1}^{n-1}\sqrt{n \cdot \left(\frac{m_s^2}{s} + \frac{m_{s+1}^2}{n-s}\right)}\right)^2 =: D^*, \quad (4)$$

where the equality is achieved if and only if

$$q_s = q_s^* \propto \sqrt{n \cdot \left(\frac{m_s^2}{s} + \frac{m_{s+1}^2}{n-s}\right)}. \quad (5)$$

In particular, $D^* \in O(1)$ for all Beta Shapley values and weighted Banzhaf values, which was already proved by Li & Yu (2024b, Proposition 4).

**Theorem 3.1.** *Setting $\mathbf{q} = \mathbf{q}^*$, for every $\epsilon \leq \frac{3C\sqrt{nD^*}}{2}$, there is*

$$\mathbb{P}(\|\hat{\phi} - \phi\|_2 \geq \epsilon) \leq 2\exp\left(-\frac{T\epsilon^2}{4nD^*C^2}\right)$$

*and* $\mathbb{E}[\|\hat{\phi} - \phi\|_2^2] = \frac{nD^*\mathbb{E}[u_{\mathbf{S}}^2] - \|\varphi\|_2^2}{T},$

*where* $\mathbb{E}[u_{\mathbf{S}}^2] = \sum_{\emptyset \subsetneq S \subsetneq [n]} q_s^* \binom{n}{s}^{-1}u_S^2.$

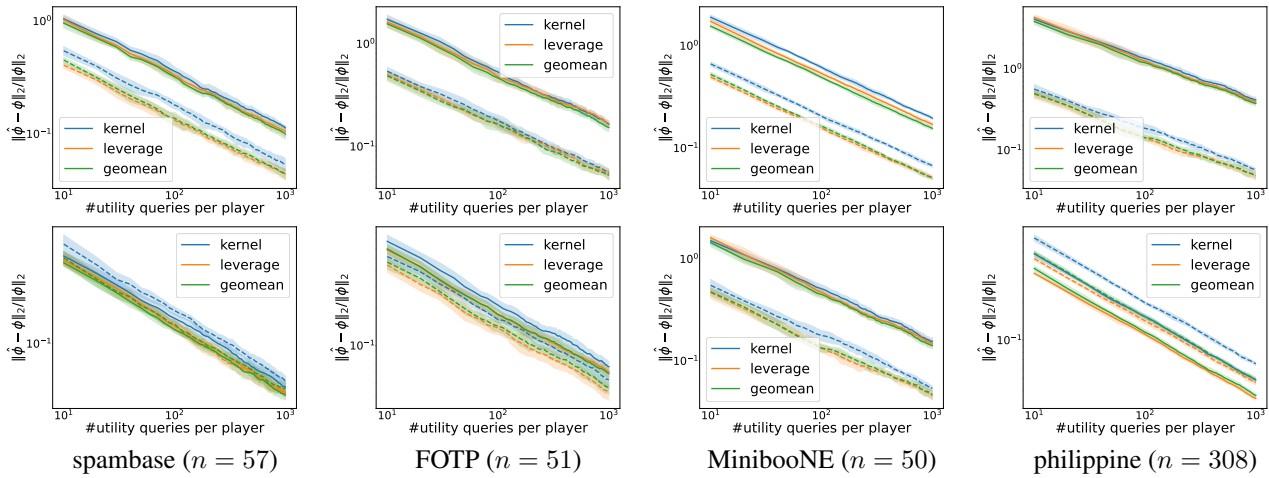

*Figure 1.* The relative approximation error of unbiased kernelSHAP in approximating the Shapley value with different sampling distributions. Here, $\lambda = \frac{u_{[n]} - u_\emptyset}{n}$. The dashed lines correspond to those with paired sampling, whereas the solid lines are without paired sampling. For the first row, the utility function there is positive, whereas for the second row, $U(S)$ can be either positive or negative.

*In particular, $\|\mathbf{z}_S\|_2^2 = nD^*$ for every $\emptyset \subsetneq S \subsetneq [n]$.*

It is worth pointing out that, in Theorem 3.1, the optimal distribution $\mathbf{q}^*$ defined in Eq. (5) uniquely satisfies two properties: (i) the expectation $\mathbb{E}[u_{\mathbf{S}}^2]$ is taken with respect to the same distribution used to approximate $\phi$ and (ii) $\|\mathbf{z}_S\|_2^2 = nD^*$ for every $S$. These properties form the foundation for designing our adaptive randomized algorithms with provably improved MSE.

**Paired sampling.** Paired sampling has become a common tool in improving the approximation quality of kernelSHAP (Covert & Lee, 2021). Empirically, however, it does not always yield performance gains (Li & Yu, 2024a), making its effectiveness somewhat mysterious. Nevertheless, our framework offers a definitive characterization.[2] Paired sampling is specific to symmetric semi-values satisfying $p_{n-s+1} = p_s$ for every $s \in [n]$, which include the Shapley value and the Banzhaf value (Banzhaf III, 1965). Instead of sampling subsets independently, paired sampling inserts the complement of each sampled subset immediately after it.

**Theorem 3.2.** *Let $T$ be the total number of utility queries, accounting for the fact that each sampled subset incurs 2 utility queries under paired sampling. Then, when $q_s = q_{n-s}$ for every $s \in [n-1]$, The use of paired sampling technique reduces to approximating $\frac{U - U^c}{2}$, where $U^c(S) := U([n] \setminus S)$. In particular,*

$$\mathbb{E}[\|\hat{\phi}^{\text{paired}} - \phi\|_2^2] = \frac{nD(\mathbf{q})\sigma_{\tilde{q}}^2 - 2\|\varphi\|_2^2}{T},$$

*where $\sigma_{\tilde{\mathbf{q}}}^2 := \left( \mathbb{E}_{\tilde{\mathbf{q}}}[u_{\mathbf{S}}^2] - \mathbb{E}_{\tilde{\mathbf{q}}}[u_{\mathbf{S}} \cdot u_{[n] \setminus \mathbf{S}}] \right)$.*

---

[2]Concurrently, Fumagalli et al. (2026) also shed light on the mechanism of paired sampling in least-square-based randomized algorithms.

Note that for $\mathbf{q}^*$ in Eq. (5), indeed $q_s^* = q_{n-s}^*$ for all $s$. Without using paired sampling, the corresponding MSE is

$$\mathbb{E}[\|\hat{\phi} - \phi\|_2^2] = \frac{nD(\mathbf{q})\mathbb{E}_{\tilde{\mathbf{q}}}[u_{\mathbf{S}}^2] - \|\varphi\|_2^2}{T}.$$

Therefore, the use of paired sampling improves the MSE if and only if

$$nD(\mathbf{q})\mathbb{E}_{\tilde{\mathbf{q}}}[u_{\mathbf{S}} \cdot u_{[n] \setminus \mathbf{S}}] + \|\varphi\|_2^2 > 0.$$

Clearly, this improvement occurs when $\mathbb{E}_{\tilde{\mathbf{q}}}[u_{\mathbf{S}} \cdot u_{[n] \setminus \mathbf{S}}] > 0$, which holds whenever $U$ does not change sign. As shown in Figure 1, Theorem 3.2 exactly predicts when paired sampling boosts the approximation. In particular, when $\mathbb{E}_{\tilde{\mathbf{q}}}[u_{\mathbf{S}} \cdot u_{[n] \setminus \mathbf{S}}] > 0$ is not satisfied, paired sampling can even degrade performance.

Next, we demonstrate how our framework bridges the existing approaches. More details can be found in Appendix B.

**Unbiased KernelSHAP.** For the Shapley value, $m_s = \frac{1}{n}$ for every $s$ and the optimal sampling distribution of our framework is $q_s \propto \sqrt{\frac{1}{s(n-s)}}$, as defined in Eq. (5). Very recently, Chen et al. (2025) established a provable framework that unifies all kernelSHAP variants to approximate the Shapley value. In particular, their unified formula for unbiased kernelSHAP can be simplified as

$$\hat{\phi}^{\text{kernel}} := \frac{1}{T} \sum_{t=1}^{T} (u_{S_t} - \lambda \cdot s_t)\mathbf{z}_{S_t} + \frac{U([n]) - U(\emptyset)}{n}\mathbf{1}_n.$$

Here, $\lambda \in \mathbb{R}$ is arbitrary. Within our framework, the arbitrariness of $\lambda$ can be directly generalized.

**Lemma 3.3.** *For the Shapley value, let $V$ be any utility function such that $V(S) = f(s)$ for every $S$. Then $\mathbb{E}[v_{\mathbf{S}}\mathbf{z}_{\mathbf{S}}] = \mathbf{0}_n$, where $v_S = V(S)$.*

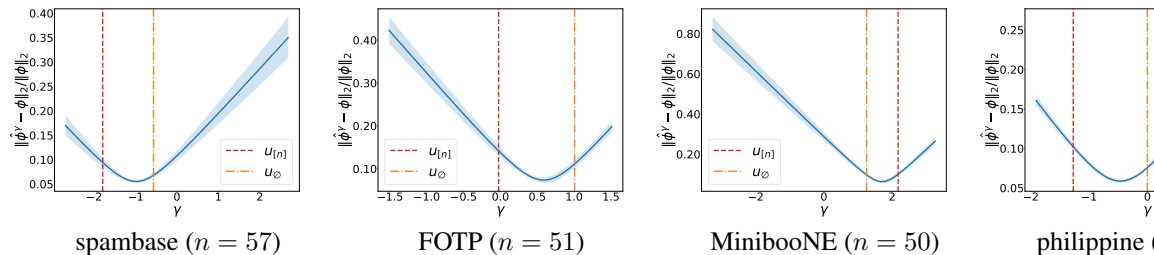

*Figure 2.* The relative approximation error of $\hat{\phi}^\gamma$ in approximating the Shapley value as $\gamma$ varies.

As a result, $\lambda \cdot s_t$ in unbiased kernelSHAP can be replaced by $f(s_t)$.

For the vanilla unbiased KernelSHAP (Covert & Lee, 2021), the sampling distribution satisfies $q_s \propto \frac{1}{s(n-s)}$ with $\lambda = 0$, whereas the leverage score sampling uses $q_s = \frac{1}{n-1}$ and $\lambda = \frac{U([n])-U(\emptyset)}{n}$ (Musco & Witter, 2025; Chen et al., 2025). Subsequently, Chen et al. (2025) proposed a modified variant by setting $\mathbf{q}$ to the geometric mean of these two distributions, which coincides with $\mathbf{q}^*$. Therefore, by Theorem 2.1, the query complexity of the modified unbiased kernelSHAP is already $O\left(\frac{n}{\epsilon^2} \log \frac{1}{\delta}\right)$. By contrast, the other two methods incur an additional multiplicative factor of $\log n$.

**Corollary 3.4.** *To ensure $\mathbb{P}(\|\hat{\phi} - \phi\|_2 \geq \epsilon) \leq \delta$, the unbiased kernelSHAP using leverage score sampling requires at most $\frac{72C^2 n \log n}{\epsilon^2} \log \frac{2}{\delta}$ utility queries for $\epsilon \leq 4C \log n$, whereas the vanilla unbiased kernelSHAP requires $\frac{8C^2 n \log n}{\epsilon^2} \log \frac{2}{\delta}$ for $\epsilon \leq Cn^{\frac{1}{2}}$. In other words, their query complexities are both $O\left(\frac{n \log n}{\epsilon^2} \log \frac{1}{\delta}\right)$.*

This suggests that the modified unbiased KernelSHAP is superior in terms of query complexity.

*Remark.* We notice that Chen et al. (2025, Proposition E.1) constructed a specific utility function to show that the modified unbiased kernelSHAP could behave worse by a multiplicative factor of $\sqrt{n}$ compared to the variant using leverage score sampling. This does not contradict our Corollary 3.4, since their constructed $U$ satisfies $\|U\|_\infty \in \Theta(n)$, whereas our query complexity analysis assumes $\|U\|_\infty \leq C$.

**SHAP-IQ.** As a weighted extension of kernelSHAP for approximating semi-values (Fumagalli et al., 2024), the estimate of SHAP-IQ can be rewritten as:

$$\hat{\phi}^{\mathrm{IQ}} := \frac{1}{T} \sum_{t=1}^{T} (u_{S_t} - u_\emptyset) \mathbf{z}_{S_t} + m_n \cdot (u_{[n]} - u_\emptyset)$$

with the sampling distribution $q_s \propto \frac{1}{s(n-s)}$. It also fits into our framework, by simply translating $\{u_S\}_S$ to $\{u_S - u_\emptyset\}_S$. Therefore, as indicated by Eq. (4), the choice of $\mathbf{q}$ in SHAP-IQ does not minimize the query complexity. As shown in

Figure 3, such non-optimality does affect the approximation performance.

**Regression-adjusted approach.** Recently, Witter et al. (2025) proposed learning a utility function $V$, whose semi-values can be computed in polynomial time, by minimizing $\mathbb{E}[\|\hat{\phi}^{\mathrm{MSR}}(U - V)\|_2^2]$. The regression-adjusted estimate is then computed as

$$\hat{\phi}^{\mathrm{adjusted}}(U) := \hat{\phi}^{\mathrm{MSR}}(U - V) + \phi(V).$$

Clearly, this approach aims to minimize the MSE. Adopting the maximum-sample-reuse (MSR) perspective with $\overline{\mathbf{q}} \in \mathbb{R}^{n+1}$ unspecified, Witter et al. (2025) derived an exact expression for $\mathbb{E}[\|\hat{\phi}(U - V)\|_2^2]$. They then heuristically choose $\overline{\mathbf{q}}$ so that no reweighting scheme is required to approximate this quantity. Notably, this choice coincides with the optimal solution for minimizing the query complexity. Specifically, the choice of their $\overline{\mathbf{q}}^{MSR} \in \mathbb{R}^{n+1}$ can be simplified as

$$\overline{q}_s^{\mathrm{MSR}} \propto \sqrt{\frac{m_s^2}{s} + \frac{m_{s+1}^2}{n-s}} \quad \text{for every } 0 \leq s \leq n,$$

where the convention here is $\frac{x}{0} := 0$. The difference is that they include $[n]$ and $\emptyset$ in the sampling pool, whereas we exclude them. In particular, Theorem 3.1 continues to hold when using $\overline{\mathbf{q}}^{\mathrm{MSR}}$ (see Appendix B), indicating that a linear-time, linear-space randomized algorithm already exists for approximating semi-values.

## 4. Our Adaptive Randomized Algorithms for Approximating Semi-Values

In this section, by applying the well-known control variates technique, we show that there is still room for improving the existing linear-time, linear-space randomized algorithms through minimizing the MSE. From now on, we assume $\mathbf{q}^*$ is employed, as it optimizes the query complexity.

Since $\phi(U) = \mathbf{0}_n$ if $U$ is constant, we immediately have the following unbiased estimate for $\phi$,

$$\hat{\phi}^\gamma := \frac{1}{T} \sum_{t=1}^{T} (u_{S_t} - \gamma) \mathbf{z}_{S_t} + \mathbf{b}$$

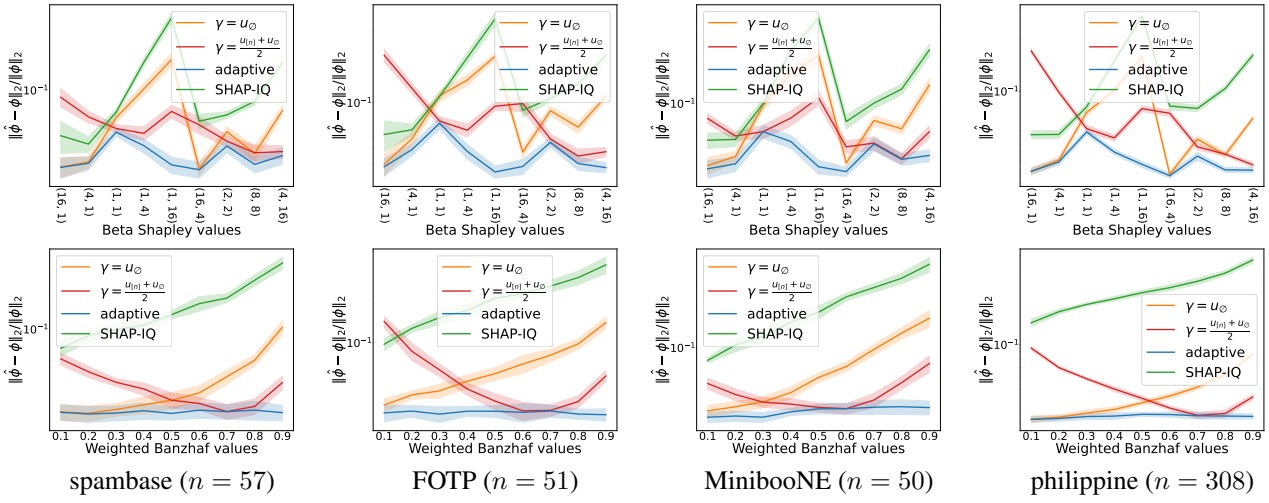

*Figure 3.* The relative approximation error of SHAP-IQ, $\hat{\boldsymbol{\phi}}^{\gamma}$ and its adaptive variant. The adaptive method corresponds to Adalina, presented in Algorithm 1, which adaptively estimates the optimal $\gamma$. Weighted Banzhaf values are parameterized by $w \in (0, 1)$, whereas Beta Shapley values are parameterized by $(\alpha, \beta)$, with $(1, 1)$ corresponding to the Shapley value.

---

**Algorithm 1** Adalina (**Ada**ptive **Lin**ear **A**pproximation)

**Input:** Weight vector $\mathbf{m} \in \mathbb{R}^n$ for semi-value $\phi$, total number of samples $T$

**Output:** Estimate $\hat{\boldsymbol{\phi}}^{\text{Adalina}}$

1 Compute the sampling distribution $\mathbf{q} \in \mathbb{R}^{n-1}$ such that
$$q_s \propto \sqrt{\frac{m_s^2}{s} + \frac{m_{s+1}^2}{n-s}}$$
2 Initialize $\hat{\boldsymbol{\varphi}}, \hat{\mathbf{v}} \leftarrow \mathbf{0}_n, \hat{\gamma} \leftarrow 0$
3 **for** $t = 1, 2, \ldots, T$ **do**
4      Sample a subset size $s$ with probability $q_s$
5      Sample a subset $S$ of size $s$ uniformly from $2^{[n]}$
6      $\hat{\boldsymbol{\varphi}} \leftarrow (1 - \frac{1}{t}) \cdot \hat{\boldsymbol{\varphi}} + \frac{1}{t} \cdot u_S \mathbf{z}_S$
7      $\hat{\mathbf{v}} \leftarrow (1 - \frac{1}{t}) \cdot \hat{\mathbf{v}} + \frac{1}{t} \cdot \mathbf{z}_S$
8      $\hat{\gamma} \leftarrow (1 - \frac{1}{t}) \cdot \hat{\gamma} + \frac{1}{t} \cdot u_S$
9 $\hat{\boldsymbol{\phi}}^{\text{Adalina}} \leftarrow \hat{\boldsymbol{\varphi}} - \hat{\gamma}\hat{\mathbf{v}} + m_n(u_{[n]} - \hat{\gamma}) - m_1(u_\emptyset - \hat{\gamma})$

---

where $\mathbf{b} := [m_n(u_{[n]} - \gamma) - m_1(u_\emptyset - \gamma)]\mathbf{1}_n$. Observe that this reduces to SHAP-IQ when $q_s \propto 1/s(n-s)$ and $\gamma = u_\emptyset$. If $m_1 = m_n$, which is satisfied by symmetric semi-values, then by Theorem 3.1,

$$\mathbb{E}[\|\hat{\boldsymbol{\phi}}^{\gamma} - \boldsymbol{\phi}\|_2^2] = \frac{nD^*\mathbb{E}[(u_{\mathbf{S}} - \gamma)^2] - \|\boldsymbol{\varphi}\|_2^2}{T}.$$

This indicates that $\gamma$ adjusts the MSE. Empirically, this is confirmed in Figure 2. In particular, the shapes of the curves align well with $\mathbb{E}[(u_S - \gamma)^2]$, indicating that there exists a unique optimal $\gamma^*$ that minimizes it. Theoretically, $\gamma^* = \mathbb{E}[u_S]$. By Theorem 3.1, the distribution underlying $\mathbb{E}[u_S^2]$ is the same as that used to approximate $\phi$, which means we can approximate $\gamma^*$ and $\phi$ simultaneously. This leads to our adaptive randomized algorithm, Adalina, for

approximating semi-values:

$$\hat{\boldsymbol{\phi}}^{\text{Adalina}} := \frac{1}{T}\sum_{t=1}^{T}(u_{S_t} - \hat{\gamma})\mathbf{z}_{S_t} + \hat{\mathbf{b}},$$

where $\hat{\mathbf{b}} := [m_n(u_{[n]} - \hat{\gamma}) - m_1(u_\emptyset - \hat{\gamma})]\mathbf{1}_n$ and $\hat{\gamma} := \frac{1}{T}\sum_{t=1}^{T} u_{S_t}$. Its procedure is summarized in Algorithm 1. Note that Adalina consumes $\Theta(n)$ space. In particular, it comes with the following improved MSE.

**Theorem 4.1.** *For semi-values satisfying $m_1 = m_n$, which include the Shapley value and the Banzhaf value, we have*

$$\mathbb{E}[\|\hat{\boldsymbol{\phi}}^{\text{Adalina}} - \boldsymbol{\phi}\|_2^2]$$
$$\leq \frac{1}{T}\left(nD^*\mathbb{E}[(u_{\mathbf{S}} - \gamma^*)^2] - \|\boldsymbol{\varphi}\|_2^2\right) + \frac{6nD^*\|U\|_\infty^2}{T(T-1)}$$
$$= \mathbb{E}[\|\hat{\boldsymbol{\phi}}^{\gamma^*} - \boldsymbol{\phi}\|_2^2] + \frac{6nD^*\|U\|_\infty^2}{T(T-1)}.$$

*For every $0 < \epsilon \leq 2C$, it requires $\frac{36nD^*C^2}{\epsilon^2}\log\frac{4}{\delta}$ utility evaluations to achieve $P(\|\hat{\boldsymbol{\phi}}^{\text{Adalina}} - \boldsymbol{\phi}\|_2 \geq \epsilon) \leq \delta$. In other words, the query complexity of $\hat{\boldsymbol{\phi}}^{\text{Adalina}}$ is $O(\frac{n}{\epsilon^2}\log\frac{1}{\delta})$ for semi-values with $D^* \in O(1)$.*

As the baseline, $\mathbb{E}[\|\hat{\boldsymbol{\phi}} - \boldsymbol{\phi}\|_2^2] = \frac{1}{T}\left(nD^*\mathbb{E}[u_{\mathbf{S}}^2] - \|\boldsymbol{\varphi}\|_2^2\right)$, which is stated in Theorem 3.1. It follows that the MSE of $\hat{\boldsymbol{\phi}}^{\text{Adalina}}$ is fast approaching that of $\hat{\boldsymbol{\phi}}^{\gamma^*}$ as $T \to \infty$; see Figure 3 for its performance.

We note that our proof relies on the condition $\mathbb{E}[\mathbf{z}_{\mathbf{S}}] = \mathbf{0}_n$, which clearly holds when $m_1 = m_n$ according to Eq. (6). This assumption can be immediately removed if $\bar{q}^{\text{MSR}}$ is used instead, since $\phi(U) = \mathbf{0}_n$ whenever $U$ is constant. In

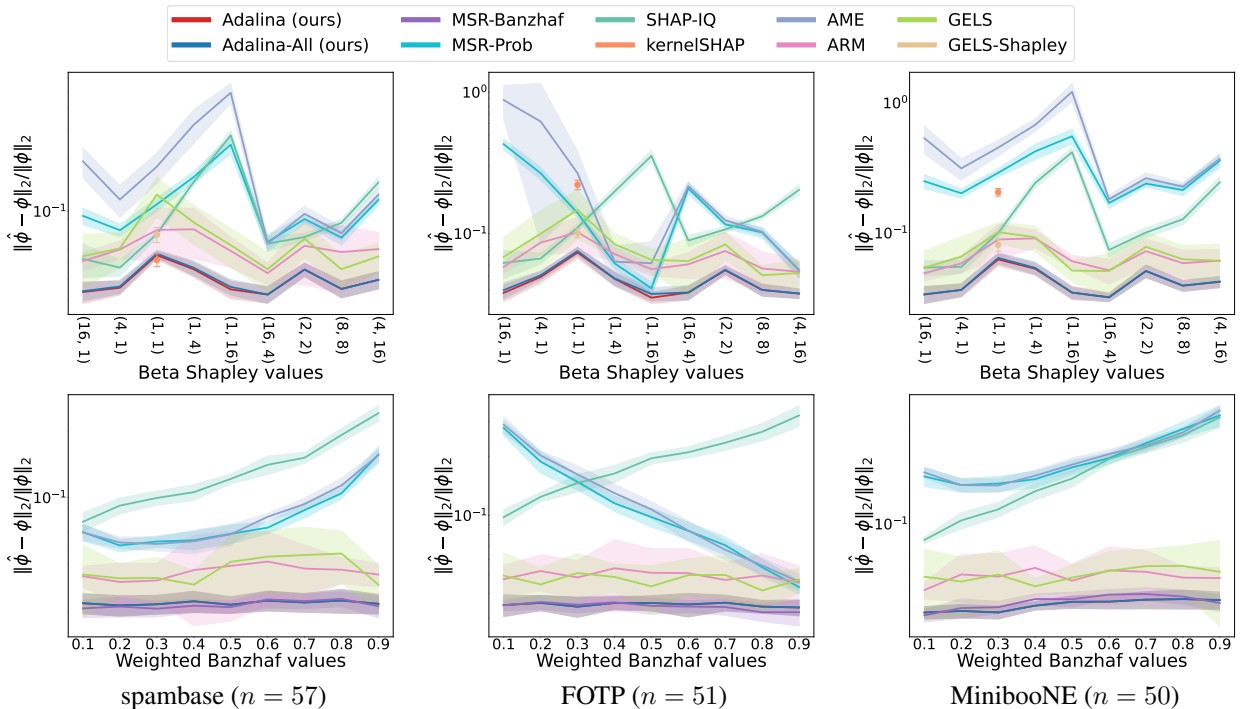

*Figure 4.* The relative approximation error of different randomized algorithms. Weighted Banzhaf values are parameterized by $w \in (0, 1)$, whereas Beta Shapley values are parameterized by $(\alpha, \beta)$, with $(1, 1)$ corresponding to the Shapley value.

this case, Theorems 3.1 and 4.1 remain valid with $\|\varphi\|_2^2$ and $D^*$ replaced by $\|\phi\|_2^2$ and $D^{\text{MSR}}$, respectively. Further details are provided in Appendix B, with the corresponding procedure summarized in Algorithm 2 (Adalina-All).

**Adalina with paired sampling.** As proved in Theorem 4.1, the MSE of $\hat{\phi}^{\text{Adalina}}$ matches that of $\hat{\phi}^{\gamma^*}$ up to a lower-order term of order $\frac{1}{T^2}$. Therefore, our analysis of $\hat{\phi}^{\gamma^*}$ naturally carries over to $\hat{\phi}^{\text{Adalina}}$. According to Theorem 3.2, while using paired sampling,[3] the MSE of $\hat{\phi}^{\gamma^*}$ is

$$\frac{nD^*(\text{Var}[u_{\mathbf{S}}] - \text{Cov}(u_{\mathbf{S}}, u_{[n] \setminus \mathbf{S}})) - 2\|\varphi\|_2^2}{T}.$$

Consequently, the MSE improves if and only if

$$nD^* \text{Cov}(u_{\mathbf{S}}, u_{[n] \setminus \mathbf{S}}) + \|\varphi\|_2^2 > 0.$$

However, if we naively apply paired sampling to Algorithms 1 and 2, the resulting estimate reduces to $\hat{\phi}^0$; that is, the effects of paired sampling and adaptivity cancel each other out. Indeed, one can verify that $\hat{\mathbf{v}} = \mathbf{0}$ when paired sampling is used. Nevertheless, as shown in the proof of Theorem 3.2, applying paired sampling is equivalent to approximating $\frac{U - U^c}{2}$ instead. Therefore, to incorporate the

---

[3] Note that $\sigma_{\tilde{\mathbf{q}}}^2$ in Theorem 4.1 is invariant under translations of $\{u_S\}_S$.

effect of paired sampling into Adalina, we may simply run Adalina to approximate $\frac{U - U^c}{2}$ using half as many sampled subsets.

## 5. Empirical Results

In this section, we examine the performance of our adaptive randomized algorithm, Adalina and its variant Adalina-All.

**Baselines.** Since our focus is on the approximation quality of $\Theta(n)$-space randomized algorithms, the baselines we consider include: MSR-Banzhaf (Wang & Jia, 2023), MSR-Prob (see $\hat{\phi}^{\text{MSR}}$ in Appendix B.4), SHAP-IQ (Fumagalli et al., 2024), unbiased kernelSHAP (Chen et al., 2025), AME (Lin et al., 2022), ARM (Kolpaczki et al., 2024), and GELS and GELS-Shapley (Li & Yu, 2024a). In particular, while the original AME requires $\Theta(n^2)$ space and an additional $O(n^3)$ time for matrix inversion, we instead use its linear-space variant provided by Li & Yu (2024b). Among these baselines, MSR-Banzhaf can only approximate weighted Banzhaf values, whereas unbiased kernelSHAP and GELS-Shapley are designed specifically for approximating the Shapley value. All the others can approximate a wide range of semi-values.

**Utility functions.** Each utility function is defined using a trained (gradient boosting) decision tree $f$ and an instance $\mathbf{x} \in \mathbb{R}^n$. Specifically, $U_f^{\mathbf{x}}(S) := \mathbb{E}_{\mathbf{X}_{[n] \setminus S}}[f(\mathbf{x}_S, \mathbf{X}_{[n] \setminus S})]$.

Therefore, the number of players is equal to the number of features. We follow the path-dependent definition given in Lundberg et al. (2020, Algorithm 1). In particular, the semi-values of $U_f^{\mathbf{x}}$ can be computed in polynomial time (Yu et al., 2022; Muschalik et al., 2024), providing ground-truths for evaluating the performance of different randomized algorithms. We adopt the numerically stable version recently developed by Li et al. (2026).[4] We employ twelve datasets from OpenML for training (gradient boosting) decision trees, which are (1) spambase ($n = 57$), (2) FOTP ($n = 51$) (Bridge et al., 2014), (3) MinibooNE ($n = 50$) (Roe et al., 2005), (4) philippine ($n = 308$), (5) GPSP ($n = 32$) (Madeo et al., 2013), (6) jannis ($n = 54$), (7) supperconduct ($n = 81$), (8) wave_energy ($n = 48$), (9) BT ($n = 77$) (Kawala et al., 2013), (10) har ($n = 561$) (Anguita et al., 2012), (11) Fashion-MNIST ($n = 784$) (Xiao et al., 2017), and (12) CIFAR_small ($n = 3,072$) (Krizhevsky et al., 2009). All gradient boosting decision trees are trained using GradientBoostingClassifier and GradientBoostingRegressor from the scikit-learn library (Pedregosa et al., 2011) with the number of trees set to 10.

Except for the last three datasets, each estimate is computed using $1,000$ queries per player; for example, when $n = 10$, this corresponds to $10,000$ queries per estimate. This quantity is set to 100 for har and Fashion-MNIST, and to 20 for CIFAR_small. All results are averaged over 10 random seeds, with the standard deviation reported. For reproducibility, all other sources of randomness are fixed to 2026. More details and experimental results are in Appendix C.

As presented in Figure 4, except for the Shapley value, i.e., the Beta Shapley value with parameter $(1,1)$, our Adalina performs consistently well. Although Adalina does not improve the MSE for non-symmetric semi-values, its estimates closely match those of Adalina-All, which theoretically achieves improved MSE for all semi-values. For the Shapley value, unbiased kernelSHAP can be significantly better, suggesting the possibility that $\inf_{\lambda \in \mathbb{R}} \mathbb{E}[(u_S - \lambda \cdot s)^2] < \inf_{\gamma \in \mathbb{R}} \mathbb{E}[(u_S - \gamma)^2]$ and indicating that there is still more room to better approximate the Shapley value. In particular, Lemma 3.3 suggests a path for future work.

## 6. Conclusion

In this work, we adopt a holistic perspective to systematically establish a theoretical framework for designing linear-time, linear-space randomized algorithms that approximate semi-values. Our framework bridges the recent works, including OFA, unbiased kernelSHAP, SHAP-IQ and the regression-adjust approach, and provides sharper query complexities. It also characterizes when paired sampling boosts

performance, which is empirically verified in Figure 1. In particular, our work enables the explicit minimization of the MSE for each utility function, through which we propose the first adaptive randomized algorithm, Adalina, with provably improved MSE. Empirically, Adalina consistently performs well against baselines. We view our framework as the first concrete step towards designing more efficient adaptive randomized algorithms for semi-value estimation.

## Acknowledgment

This research is supported by the National Research Foundation Singapore and the Singapore Ministry of Digital Development and Innovation, National AI Group under the AI Visiting Professorship Programme (award number AIVP-2024-001). YY gratefully acknowledges NSERC and CIFAR for funding support.

## Impact Statement

This paper presents work whose goal is to advance the field of machine learning. There are many potential societal consequences of our work, none of which we feel must be specifically highlighted here.

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

# A. Proofs

**Theorem A.1** (Yurinsky, 1995, Theorem 3.3.4). *Let* $\mathbf{S} = \sum_{i=1}^{M} \mathbf{X}_i$ *where* $\{\mathbf{X}_i\}_{i=1}^{M}$ *are independent zero-mean random vectors. Then, for every* $\lambda > 0$,

$$\mathbb{E}[\cosh(\lambda\|\mathbf{S}\|_2)] \leq \prod_{i=1}^{M} \mathbb{E}[\exp(\lambda\|\mathbf{X}_i\|_2) - \lambda\|\mathbf{X}_i\|_2].$$

*Proof.* By Taylor expansion (and the monotone convergence theorem),

$$\mathbb{E}[\cosh(\lambda\|\mathbf{S}\|_2)] = \sum_{\ell=0}^{\infty} \frac{\lambda^{2\ell}}{(2\ell)!} \mathbb{E}[\|\mathbf{S}\|_2^{2\ell}].$$

We begin by bounding each moment $\mathbb{E}[\|\mathbf{S}\|_2^{2\ell}]$. Expanding the square Euclidean norm we have

$$\|\mathbf{S}\|_2^{2\ell} = \langle \mathbf{S}, \mathbf{S} \rangle^{\ell} = \sum_{I \in [M]^{[\ell]}} \sum_{J \in [M]^{[\ell]}} \prod_{k=1}^{\ell} \langle \mathbf{X}_{I(k)}, \mathbf{X}_{J(k)} \rangle,$$

where $[M] := \{1, 2, \ldots, M\}$. For every $(I, J)$ and $i \in [M]$, define

$$\kappa_i(I, J) := \sum_{k=1}^{\ell} \left( [\![ I(k) = i ]\!] + [\![ J(k) = i ]\!] \right).$$

Then, since $\{\mathbf{X}_i\}_{i=1}^{M}$ are zero-mean, we have

$$\mathbb{E}\left[ \prod_{k=1}^{\ell} \langle \mathbf{X}_{I(k)}, \mathbf{X}_{J(k)} \rangle \right] = 0 \text{ if } \kappa_i(I, J) = 1 \text{ for some } i \in [M].$$

For the other cases,

$$\mathbb{E}\left[ \prod_{k=1}^{\ell} \langle \mathbf{X}_{I(k)}, \mathbf{X}_{J(k)} \rangle \right] \leq \mathbb{E}\left[ \prod_{k=1}^{\ell} \left( \|\mathbf{X}_{I(k)}\|_2 \cdot \|\mathbf{X}_{J(k)}\|_2 \right) \right] = \mathbb{E}\left[ \prod_{i=1}^{M} \|\mathbf{X}_i\|_2^{\kappa_i(I,J)} \right].$$

Let

$$K_\ell := \{ \kappa = (\kappa_i) \in \mathbb{Z}_+^M \mid \sum_{i=1}^{M} \kappa_i = 2\ell \text{ and } \kappa_i \neq 1 \text{ for every } i \in [M] \}.$$

Then, rearranging the sum and applying the above inequality, we have

$$\mathbb{E}\left[ \|\mathbf{S}\|_2^{2\ell} \right] \leq \sum_{\kappa \in K_\ell} \sum_{(I,J) : \kappa(I,J)=\kappa} \mathbb{E}\left[ \prod_{i=1}^{M} \|\mathbf{X}_i\|_2^{\kappa_i} \right].$$

For each $\kappa \in K_\ell$, there are $2\ell$ entries in $(I, J)$ to be determined, with the constraint that $\kappa_i$ entries are chosen to be $i \in [M]$. It follows that

$$|\{(I, J) : \kappa(I, J) = \kappa\}| = \frac{(2\ell)!}{\prod_{i=1}^{M} \kappa_i!}.$$

Consequently,

$$\mathbb{E}\left[ \|\mathbf{S}\|_2^{2\ell} \right] \leq \sum_{\kappa \in K_\ell} (2\ell)! \mathbb{E}\left[ \prod_{i=1}^{M} \frac{\|\mathbf{X}_i\|_2^{\kappa_i}}{\kappa_i!} \right] = \sum_{\kappa \in K_\ell} (2\ell)! \prod_{i=1}^{M} \frac{\mathbb{E}[\|\mathbf{X}_i\|_2^{\kappa_i}]}{\kappa_i!},$$

where the equality comes from the independence among $\{\mathbf{X}_i\}_{i=1}^{M}$. As a result,

$$\mathbb{E}[\cosh(\lambda\|\mathbf{S}\|_2)] \leq \sum_{\ell=0}^{\infty} \sum_{\kappa \in K_\ell} \prod_{i=1}^{M} \frac{\lambda^{\kappa_i} \mathbb{E}[\|\mathbf{X}_i\|_2^{\kappa_i}]}{\kappa_i!}.$$

Since

$$\bigcup_{\ell=0}^{\infty} K_\ell \subseteq (\mathbb{Z}_+ \setminus \{\mathbf{1}\})^M,$$

we eventually have

$$\mathbb{E}[\cosh(\lambda\|\mathbf{S}\|_2)] \le \sum_{\kappa_1 \neq 1} \cdots \sum_{\kappa_M \neq 1} \prod_{i=1}^{M} \frac{\lambda^{\kappa_i}\mathbb{E}[\|\mathbf{X}_i\|_2^{\kappa_i}]}{\kappa_i!} = \prod_{i=1}^{M} \sum_{\kappa_i \neq 1} \frac{\lambda^{\kappa_i}\mathbb{E}[\|\mathbf{X}_i\|_2^{\kappa_i}]}{\kappa_i!} = \prod_{i=1}^{M} \mathbb{E}[\exp(\lambda\|\mathbf{X}_i\|_2) - \lambda\|\mathbf{X}_i\|_2],$$

which completes the proof. $\qquad\square$

**Theorem 2.1** (Vector concentration inequality). *Suppose $\{\mathbf{X}_i\}_{i=1}^{M}$ are i.i.d. zero-mean random vectors such that $\mathbb{E}[\|\mathbf{X}_i\|_2^2] \le \sigma^2$ and $\|\mathbf{X}_i\|_2 \le C$ almost surely. Then, for every $0 < \epsilon \le \frac{3\sigma^2}{C}$, there is*

$$\mathbb{P}\left(\left\|\frac{1}{M}\sum_{i=1}^{M} \mathbf{X}_i\right\|_2 \ge \epsilon\right) \le 2\exp\left(-\frac{M\epsilon^2}{4\sigma^2}\right).$$

*Proof.* The proof presented here is routine in establishing Bernstein's inequality. Let $\mathbf{S} := \sum_{i=1}^{M} \mathbf{X}_i$ and $\lambda > 0$, and we begin by applying Markov's inequality to obtain

$$\mathbb{P}\left(\|\mathbf{S}\|_2 \ge \epsilon\right) = \mathbb{P}\left(\cosh(\lambda\|\mathbf{S}\|_2) \ge \cosh(\lambda\epsilon)\right) \le \frac{\mathbb{E}[\cosh(\lambda\|\mathbf{S}\|_2)]}{\cosh(\lambda\epsilon)}.$$

Then, by Theorem A.1,

$$\mathbb{E}[\cosh(\lambda\|\mathbf{S}\|_2)] \le \left(\mathbb{E}\left[\exp\left(\lambda\|\mathbf{X}_i\|_2 - \lambda\|\mathbf{X}_i\|_2\right)\right]\right)^M.$$

Introducing the non-decreasing and non-negative function $h(x) := \frac{e^x - 1 - x}{x^2}$,

$$\mathbb{E}[\exp(\lambda\|\mathbf{X}_i\|_2) - \lambda\|\mathbf{X}_i\|_2] = 1 + \mathbb{E}[\lambda^2\|\mathbf{X}_i\|_2^2 \cdot h(\lambda\|\mathbf{X}_i\|_2)] \le 1 + \lambda^2\sigma^2 \cdot h(\lambda C).$$

We continue to bound $h$ by Taylor expansion (and the simple fact that $k! \ge 1 \cdot 2 \cdot 3^{k-2}$):

$$h(\lambda C) = \sum_{k=2}^{\infty} \frac{(\lambda C)^{k-2}}{k!} \le \frac{1}{2}\sum_{k=2}^{\infty} \frac{(\lambda C)^{k-2}}{3^{k-2}} = \frac{1}{2}\sum_{k=0}^{\infty} \left(\frac{\lambda C}{3}\right)^k.$$

If $\lambda < \frac{3}{C}$, by combining the previous two inequalities, we have

$$\mathbb{E}[\exp(\lambda\|\mathbf{X}_i\|_2) - \lambda\|\mathbf{X}_i\|_2] \le 1 + \frac{\lambda^2\sigma^2}{2(1 - \frac{C}{3}\lambda)} \le \exp\left(\frac{\lambda^2\sigma^2}{2(1 - \frac{C}{3}\lambda)}\right),$$

where the last inequality comes from $1 + x \le e^x$. As a result,

$$\mathbb{E}[\cosh(\lambda\|\mathbf{S}\|_2)] \le \exp\left(\frac{M\lambda^2\sigma^2}{2(1 - \frac{C}{3}\lambda)}\right).$$

Since $\cosh(x) \ge \frac{e^x}{2}$,

$$\mathbb{P}(\|\mathbf{S}\|_2 \ge \epsilon) \le 2\exp\left(\frac{M\lambda^2\sigma^2}{2(1 - \frac{C}{3}\lambda)} - \lambda\epsilon\right).$$

Putting $\lambda = \frac{\epsilon}{M\sigma^2 + \frac{C}{3}\epsilon}$, which meets the constraint $\lambda < \frac{3}{C}$, we have

$$\mathbb{P}(\|\mathbf{S}\|_2 \ge \epsilon) \le 2\exp\left(-\frac{\epsilon^2}{2(M\sigma^2 + \frac{C}{3}\epsilon)}\right),$$

which is equivalent to $\mathbb{P}(\|\frac{1}{M}\sum_{i=1}^{M} \mathbf{X}_i\|_2 \ge \epsilon) \le 2\exp\left(-\frac{M\epsilon^2}{2(\sigma^2 + \frac{C}{3}\epsilon)}\right)$. If $\frac{C}{3}\epsilon \le \sigma^2$, which is equivalent to $\epsilon \le \frac{3\sigma^2}{C}$, the result follows. $\qquad\square$

**Theorem 3.1.** *Setting* $\mathbf{q} = \mathbf{q}^*$, *for every* $\epsilon \leq \frac{3C\sqrt{nD^*}}{2}$, *there is*

$$\mathbb{P}(\|\hat{\boldsymbol{\phi}} - \boldsymbol{\phi}\|_2 \geq \epsilon) \leq 2\exp\left(-\frac{T\epsilon^2}{4nD^*C^2}\right)$$

$$\text{and } \mathbb{E}[\|\hat{\boldsymbol{\phi}} - \boldsymbol{\phi}\|_2^2] = \frac{nD^*\mathbb{E}[u_{\mathbf{S}}^2] - \|\boldsymbol{\varphi}\|_2^2}{T},$$

$$\text{where } \mathbb{E}[u_{\mathbf{S}}^2] = \sum_{\emptyset \subsetneq S \subsetneq [n]} q_s^* \binom{n}{s}^{-1} u_S^2.$$

*In particular,* $\|\mathbf{z}_S\|_2^2 = nD^*$ *for every* $\emptyset \subsetneq S \subsetneq [n]$.

*Proof.* Recall that $D(\mathbf{q}) = \sum_{s=1}^{n-1} \frac{n}{q_s}\left(\frac{m_s^2}{s} + \frac{m_{s+1}^2}{n-s}\right)$. Let $A := \sum_{s=1}^{n-1} \sqrt{n\left(\frac{m_s^2}{s} + \frac{m_{s+1}^2}{n-s}\right)}$. Then, the unique optimal choice of $\mathbf{q}$ can be written as

$$q_s = q_s^* := \frac{1}{A}\sqrt{n\left(\frac{m_s^2}{s} + \frac{m_{s+1}^2}{n-s}\right)}.$$

In particular,

$$D^* = A \cdot \sum_{s=1}^{n-1} \sqrt{n\left(\frac{m_s^2}{s} + \frac{m_{s+1}^2}{n-s}\right)} = A^2.$$

For any $\mathbf{z}_S$,

$$\|\mathbf{z}_S\|_2^2 = \frac{n^2}{q_s^2}\left(\frac{m_s^2}{s} + \frac{m_{s+1}^2}{n-s}\right) = n \cdot A^2 = n \cdot D^*.$$

Therefore, we have

$$\mathbb{E}[\|u_{\mathbf{S}}\mathbf{z}_{\mathbf{S}} - \boldsymbol{\varphi}\|_2^2] = \mathbb{E}[\|u_{\mathbf{S}}\mathbf{z}_{\mathbf{S}}\|_2^2] - \|\boldsymbol{\varphi}\|_2^2 \leq \mathbb{E}[\|u_{\mathbf{S}}\mathbf{z}_{\mathbf{S}}\|_2^2] = n \cdot D^* \cdot \sum_{\emptyset \subsetneq S \subsetneq [n]} q_s \binom{n}{s}^{-1} u_S^2$$

$$\text{and } \|u_S\mathbf{z}_S - \boldsymbol{\varphi}\|_2 \leq \|u_S\mathbf{z}_S\|_2 + \mathbb{E}[\|u_{\mathbf{S}}\mathbf{z}_{\mathbf{S}}\|_2] \leq 2C\sqrt{nD^*}.$$

Invoking Theorem 2.1 yields the desired result. $\qquad\square$

**Theorem 3.2.** *Let $T$ be the total number of utility queries, accounting for the fact that each sampled subset incurs 2 utility queries under paired sampling. Then, when $q_s = q_{n-s}$ for every $s \in [n-1]$, The use of paired sampling technique reduces to approximating $\frac{U - U^c}{2}$, where $U^c(S) := U([n] \setminus S)$. In particular,*

$$\mathbb{E}[\|\hat{\boldsymbol{\phi}}^{\text{paired}} - \boldsymbol{\phi}\|_2^2] = \frac{nD(\mathbf{q})\sigma_{\tilde{q}}^2 - 2\|\boldsymbol{\varphi}\|_2^2}{T},$$

*where* $\sigma_{\tilde{\mathbf{q}}}^2 := \left(\mathbb{E}_{\tilde{\mathbf{q}}}[u_{\mathbf{S}}^2] - \mathbb{E}_{\tilde{\mathbf{q}}}[u_{\mathbf{S}} \cdot u_{[n]\setminus\mathbf{S}}]\right)$.

*Proof.* Recall that $p_s = p_{n+1-s}$ for every $s \in [n]$ if $\phi$ corresponds to a symmetric semi-value. As a result, we have

$$m_s = m_{n+1-s} \text{ for every } s \in [n].$$

Let $T$ be the total number of utility queries. Then, after sampling $\frac{T}{2}$ subsets $R_1, R_2, \ldots, R_{\frac{T}{2}}$, the sequence of subsets used to compute $\hat{\boldsymbol{\phi}}$ is

$$S_1 = R_1, \ S_2 = [n] \setminus R_1, \ S_3 = R_2, \ S_4 = [n] \setminus R_3, \ldots, S_{T-1} = R_{\frac{T}{2}}, \ S_T = [n] \setminus R_{\frac{T}{2}}.$$

Recall that the estimate is computed as $\hat{\boldsymbol{\varphi}} = \sum_{t=1}^{T} u_{S_t}\mathbf{z}_{S_t}$, which can be rewritten as

$$\hat{\boldsymbol{\varphi}} = \frac{2}{T}\sum_{t=1}^{\frac{T}{2}} \frac{1}{2}\left(u_{R_t}\mathbf{z}_{R_t} + u_{[n]\setminus R_t}\mathbf{z}_{[n]\setminus R_t}\right).$$

Specifically,

$$\frac{1}{2}\left(u_{R_t}\mathbf{z}_{R_t} + u_{[n]\setminus R_t}\mathbf{z}_{[n]\setminus R_t}\right) = \frac{u_{R_t} - u_{[n]\setminus R_t}}{2}\cdot\mathbf{z}_{R_t}.$$

For symmetric semi-values, we have

$$\mathbb{E}\left[\frac{u_{\mathbf{S}} - u_{[n]\setminus\mathbf{S}}}{2}\cdot\mathbf{z}_S\right] = \frac{\boldsymbol{\varphi} + \boldsymbol{\varphi}}{2} = \boldsymbol{\varphi}.$$

Then,

$$\mathbb{E}[\|\hat{\boldsymbol{\phi}} - \boldsymbol{\phi}\|_2^2] = \frac{2\left(\mathbb{E}[\|\frac{u_{\mathbf{S}} - u_{[n]\setminus\mathbf{S}}}{2}\cdot\mathbf{z}_{\mathbf{S}}\|_2^2] - \|\boldsymbol{\varphi}\|_2^2\right)}{T} = \frac{n\cdot D(\mathbf{q})\cdot\mathbb{E}_{\tilde{\mathbf{q}}}[\frac{(u_{\mathbf{S}} - u_{[n]\setminus\mathbf{S}})^2}{2}] - 2\|\boldsymbol{\varphi}\|_2^2}{T}.$$

Since $\mathbb{E}_{\tilde{\mathbf{q}}}[u_{\mathbf{S}}^2] = \mathbb{E}_{\tilde{\mathbf{q}}}[u_{[n]\setminus\mathbf{S}}^2]$,

$$\mathbb{E}_{\tilde{\mathbf{q}}}\left[\frac{(u_{\mathbf{S}} - u_{[n]\setminus\mathbf{S}})^2}{2}\right] = \mathbb{E}_{\tilde{\mathbf{q}}}[u_{\mathbf{S}}^2] - \mathbb{E}_{\tilde{\mathbf{q}}}[u_{\mathbf{S}}\cdot u_{[n]\setminus\mathbf{S}}].$$

$\square$

**Lemma 3.3.** *For the Shapley value, let $V$ be any utility function such that $V(S) = f(s)$ for every $S$. Then $\mathbb{E}[v_{\mathbf{S}}\mathbf{z}_{\mathbf{S}}] = \mathbf{0}_n$, where $v_S = V(S)$.*

*Proof.* Observe that

$$\mathbb{E}[v_{\mathbf{S}}\mathbf{z}_{\mathbf{S}}] = \mathbf{W}^\top\mathbf{v}.$$

where $\mathbf{w}_S$ is the $S$-th column of $\mathbf{W}^\top$. For the Shapley value,

$$W_{S,i} = \begin{cases} \frac{1}{n}\binom{n-1}{s-1}^{-1}, & i\in S, \\ -\frac{1}{n}\binom{n-1}{s}^{-1}, & \text{otherwise.} \end{cases}$$

Specifically,

$$\begin{aligned}
(\mathbf{W}^\top\mathbf{v})_i &= \frac{1}{n}\sum_{i\in S}\binom{n-1}{s-1}f(s) - \frac{1}{n}\sum_{i\notin S}\binom{n-1}{s}f(s) \\
&= \frac{1}{n}\sum_{s=1}^{n-1}\binom{n-1}{s-1}^{-1}\binom{n-1}{s-1}f(s) - \frac{1}{n}\sum_{s=1}^{n-1}\binom{n-1}{s}^{-1}\binom{n-1}{s}f(s) \\
&= \frac{1}{n}\sum_{s=1}^{n-1}[f(s) - f(s)] = 0.
\end{aligned}$$

$\square$

**Corollary 3.4.** *To ensure $\mathbb{P}(\|\hat{\boldsymbol{\phi}} - \boldsymbol{\phi}\|_2 \geq \epsilon) \leq \delta$, the unbiased kernelSHAP using leverage score sampling requires at most $\frac{72C^2 n\log n}{\epsilon^2}\log\frac{2}{\delta}$ utility queries for $\epsilon \leq 4C\log n$, whereas the vanilla unbiased kernelSHAP requires $\frac{8C^2 n\log n}{\epsilon^2}\log\frac{2}{\delta}$ for $\epsilon \leq Cn^{\frac{1}{2}}$. In other words, their query complexities are both $O(\frac{n\log n}{\epsilon^2}\log\frac{1}{\delta})$.*

*Proof.* Given a sequence of sampled subsets $\{S_t\}_{t=1}^T$,

$$\hat{\boldsymbol{\phi}}^{\text{leverage}} := \frac{1}{T}\sum_{t=1}^T\left(u_{S_t} - \frac{[u_{[n]} - u_{\emptyset}]s_t}{n}\right)\cdot\mathbf{z}_{S_t} + \frac{u_{[n]} - u_{\emptyset}}{n}\mathbf{1}_n.$$

Specifically,

$$(\mathbf{z}_S)_i = \frac{n}{s}[\![i\in S]\!] - \frac{n}{n-s}[\![i\notin S]\!].$$

Then, for $n > 1$,

$$\mathbb{E}\left[\left\|\left(u_{\mathbf{S}} - \frac{(u_{[n]} - u_{\emptyset})\cdot|\mathbf{S}|}{n}\right)\mathbf{z}_{\mathbf{S}}\right\|_2^2\right] <= 9C^2\mathbb{E}[\|\mathbf{z}_{\mathbf{S}}\|_2^2] = 9C^2\sum_{s=1}^{n-1}\frac{n^2}{s(n-s)} \leq 18C^2 n\log n,$$

$$\left\|\left(u_S - \frac{(u_{[n]} - u_{\emptyset})\cdot s}{n}\right)\mathbf{z}_S\right\|_2 \leq 3C\|\mathbf{z}_S\|_2 = 3nC\sqrt{\frac{n}{s(n-s)}} \leq 6nC.$$

The results follow by applying Theorem 2.1. For the vanilla kernelSHAP,

$$\hat{\phi}^{\text{vanilla}} := \frac{1}{T}\sum_{t=1}^{T} u_{S_t}\mathbf{z}_{S_t} + \frac{u_{[n]} - u_{\emptyset}}{n}\mathbf{1}_n \quad \text{where} \quad \mathbf{z}_S = \begin{cases} \frac{2H_{n-1}}{n}(n-s), & i \in S, \\ -\frac{2H_{n-1}}{n}s, & \text{otherwise.} \end{cases}$$

Here, $H_{n-1} = \sum_{k=1}^{n-1}\frac{1}{k} \le \log n$. Then,

$$\mathbb{E}[\|u_{\mathbf{S}}\mathbf{z}_{\mathbf{S}}\|_2^2] \le C^2\mathbb{E}[\|\mathbf{z}_{\mathbf{S}}\|_2^2] = C^2 \cdot 2H_{n-1}(n-1) \le 2C^2 n\log n,$$

$$\|u_S\mathbf{z}_S\|_2 \le C\|\mathbf{z}_S\|_2 = \frac{2CH_{n-1}}{n}\sqrt{n(n-s)n} \le 2Cn^{\frac{1}{2}}\log n.$$

$\square$

**Lemma A.2.** *For $\mathbf{z}_{\mathbf{S}}$ defined in §3, we always have*

$$\mathbb{E}[\mathbf{z}_{\mathbf{S}}] = \mathbf{W}^\mathsf{T}\mathbf{1}_{2^n-2} = (m_1 - m_n)\cdot\mathbf{1}_n, \tag{6}$$

*where $\mathbf{w}_S$ is the $S$-th column of $\mathbf{W}^\mathsf{T}$.*

*Proof.* If $U(S)$ is constant for all $S$, it is straightforward to see from Eq. (1) that $\phi(U) = \mathbf{0}_n$. This implies that $\mathbf{W}^\mathsf{T}\mathbf{1}_{2^n-2} = (m_1 - m_n)\cdot\mathbf{1}_n$. $\square$

**Theorem 4.1.** *For semi-values satisfying $m_1 = m_n$, which include the Shapley value and the Banzhaf value, we have*

$$\mathbb{E}[\|\hat{\phi}^{\text{Adalina}} - \phi\|_2^2]$$
$$\le \frac{1}{T}\left(nD^*\mathbb{E}[(u_{\mathbf{S}} - \gamma^*)^2] - \|\varphi\|_2^2\right) + \frac{6nD^*\|U\|_\infty^2}{T(T-1)}$$
$$= \mathbb{E}[\|\hat{\phi}^{\gamma^*} - \phi\|_2^2] + \frac{6nD^*\|U\|_\infty^2}{T(T-1)}.$$

*For every $0 < \epsilon \le 2C$, it requires $\frac{36nD^*C^2}{\epsilon^2}\log\frac{4}{\delta}$ utility evaluations to achieve $P(\|\hat{\phi}^{\text{Adalina}} - \phi\|_2 \ge \epsilon) \le \delta$. In other words, the query complexity of $\hat{\phi}^{\text{Adalina}}$ is $O(\frac{n}{\epsilon^2}\log\frac{1}{\delta})$ for semi-values with $D^* \in O(1)$.*

*Proof.* For symmetric semi-values,

$$m_n(u_{[n]} - \hat{\gamma}) - m_1(u_{\emptyset} - \hat{\gamma}) = m_n u_{[n]} - m_1 u_{\emptyset}.$$

Let

$$\hat{\varphi} := \frac{1}{T}\sum_{t=1}^{T}(u_{S_t} - \hat{\gamma})\cdot\mathbf{z}_{S_t},$$

$$\mathbf{\Psi} := \sum_{t=1}^{T}\psi_t \quad \text{where} \quad \psi_t := u_{S_t}\mathbf{z}_{S_t} - \frac{1}{T}\sum_{r=1}^{T}u_{S_r}\cdot\mathbf{z}_{S_t},$$

$$\text{and} \quad \overline{\mathbf{\Psi}} := \sum_{t=1}^{T}\overline{\psi}_t \quad \text{where} \quad \overline{\psi}_t = u_{S_t}\mathbf{z}_{S_t} - \frac{1}{T-1}\sum_{r\ne t}u_{S_r}\cdot\mathbf{z}_{S_t}.$$

Then, we have

$$\frac{1}{T}\mathbf{\Psi} = \hat{\varphi} \quad \text{and} \quad \frac{1}{T}\mathbb{E}[\overline{\mathbf{\Psi}}] = \varphi.$$

Besides,

$$\psi_t - \overline{\psi}_t = \frac{1}{T(T-1)}\sum_{r\ne t}u_{S_r}\cdot\mathbf{z}_{S_t} - \frac{1}{T}u_{S_t}\mathbf{z}_{S_t} = -\frac{1}{T}\overline{\psi}_t.$$

Observe that

$$\mathbb{E}[\|\hat{\phi}^{\text{Adalina}} - \phi\|_2^2] = \mathbb{E}[\|\hat{\varphi} - \varphi\|_2^2] = \frac{1}{T^2}\mathbb{E}[\|\boldsymbol{\Psi} - \overline{\boldsymbol{\Psi}} + \overline{\boldsymbol{\Psi}} - T\cdot\varphi\|_2^2] = \frac{1}{T^2}\mathbb{E}\left[\left\|-\frac{1}{T}\overline{\boldsymbol{\Psi}} + \overline{\boldsymbol{\Psi}} - T\cdot\varphi\right\|_2^2\right].$$

Specifically,

$$\mathbb{E}\left[\left\|-\frac{1}{T}\overline{\boldsymbol{\Psi}} + \overline{\boldsymbol{\Psi}} - T\cdot\varphi\right\|_2^2\right] = \frac{1}{T^2}\mathbb{E}[\|\overline{\boldsymbol{\Psi}}\|_2^2] + \mathbb{E}[\|\overline{\boldsymbol{\Psi}} - T\cdot\varphi\|_2^2] - \frac{2}{T}\mathbb{E}[\langle\overline{\boldsymbol{\Psi}}, \overline{\boldsymbol{\Psi}} - T\cdot\varphi\rangle].$$

Since $\mathbb{E}[\overline{\boldsymbol{\Psi}}] = T\cdot\varphi$, there is $\mathbb{E}[\langle\overline{\boldsymbol{\Psi}}, \overline{\boldsymbol{\Psi}} - T\cdot\varphi\rangle] = \mathbb{E}[\|\overline{\boldsymbol{\Psi}} - T\cdot\varphi\|_2^2]$. Consequently,

$$\mathbb{E}[\|\hat{\phi}^{\text{Adalina}} - \phi\|_2^2] = \frac{1}{T^4}\mathbb{E}[\|\overline{\boldsymbol{\Psi}}\|_2^2] + \frac{T-2}{T^3}\mathbb{E}[\|\overline{\boldsymbol{\Psi}} - T\cdot\varphi\|_2^2] \le \frac{1}{T^4}\mathbb{E}[\|\overline{\boldsymbol{\Psi}}\|_2^2] + \frac{1}{T^2}\mathbb{E}[\|\overline{\boldsymbol{\Psi}} - T\cdot\varphi\|_2^2].$$

Recall that $\|\mathbf{z}_S\|_2^2 = nD^*$ for every $S$. For the first term,

$$\frac{1}{T^2}\|\overline{\boldsymbol{\Psi}}\| \le \frac{1}{T}\|\overline{\psi}_t\| \le \frac{1}{T}\sqrt{4nD^*\|U\|_\infty^2}, \text{ and thus } \frac{1}{T^4}\mathbb{E}[\|\overline{\boldsymbol{\Psi}}\|_2^2] \le \frac{4nD^*\|U\|_\infty^2}{T^2}.$$

For the other term

$$\mathbb{E}[\|\overline{\boldsymbol{\Psi}} - T\cdot\varphi\|_2^2] = T\cdot\mathbb{E}[\|\overline{\psi}_t - \varphi\|_2^2] + T(T-1)\cdot\mathbb{E}[\langle\overline{\psi}_{t_1} - \varphi, \overline{\psi}_{t_2} - \varphi\rangle] \text{ where } t_1 \ne t_2.$$

Since

$$\overline{\psi}_t - \varphi = (u_{S_t} - \gamma^*)\mathbf{z}_{S_t} + (\gamma^* - \frac{1}{T}\sum_{r\ne t}u_{S_r})\cdot\mathbf{z}_{S_t} - \varphi \text{ and } \mathbb{E}[\psi_t] = \varphi,$$

we have

$$\mathbb{E}[\|\overline{\psi}_t - \varphi\|_2^2] = nD^*\mathbb{E}[(u_\mathbf{S} - \gamma^*)^2] + \frac{nD^*}{T}\text{Var}[u_\mathbf{S}] - \|\varphi\|_2^2 \le nD^*\mathbb{E}[(u_\mathbf{S} - \gamma^*)^2] + \frac{nD^*}{T}\|U\|_\infty^2 - \|\varphi\|_2^2.$$

By Eq. 6, for symmetric semi-values, there is

$$\mathbb{E}[\mathbf{z}_\mathbf{S}] = \mathbf{0}_n.$$

As a result,

$$\mathbb{E}[\langle\overline{\psi}_{t_1} - \varphi, \overline{\psi}_{t_2} - \varphi\rangle] = \frac{1}{(T-1)^2}\|\varphi\|_2^2 \le \frac{nD^*\|U\|_\infty^2}{(T-1)^2}.$$

Combining all the results yields

$$\mathbb{E}[\|\hat{\phi}^{\text{Adalina}} - \phi\|_2^2] \le \frac{1}{T}\left(nD^*\mathbb{E}[(u_\mathbf{S} - \gamma^*)^2] - \|\varphi\|_2^2\right) + \frac{6nD^*\|U\|_\infty^2}{T(T-1)} = \mathbb{E}[\|\hat{\phi}^{\gamma^*} - \phi\|_2^2] + \frac{6nD^*\|U\|_\infty^2}{T(T-1)}.$$

Next, we prove the asymptotic complexity of $\hat{\phi}^{\text{Adalina}}$. Using Hoeffding's inequality, with probability at least $1 - \frac{\delta}{2}$, there is

$$|\hat{\gamma} - \gamma^*| < \sqrt{\frac{2C^2}{T}\log\frac{4}{\delta}}.$$

Meanwhile, according to Theorem 3.1, with probability at least $1 - \frac{\delta}{2}$,

$$\|\hat{\phi}^{\gamma^*} - \phi\| < \sqrt{\frac{16nD^*C^2}{T}\log\frac{4}{\delta}}.$$

Here, the induced constraint $\epsilon \le 3C\sqrt{nD^*}$ is equivalent to $T \ge \frac{16}{9}\log\frac{4}{\delta}$. Therefore, with probability at least $1 - \delta$, we have both inequalities. Then,

$$\|\hat{\phi}^{\text{Adalina}} - \phi\| \le \|\hat{\phi}^{\text{Adalina}} - \hat{\phi}^{\gamma^*}\| + \|\hat{\phi}^{\gamma^*} - \phi\| < \sqrt{\frac{2nD^*C^2}{T}\log\frac{4}{\delta}} + \sqrt{\frac{16nD^*C^2}{T}\log\frac{4}{\delta}} \le \sqrt{\frac{36nD^*C^2}{T}\log\frac{4}{\delta}}.$$

As a result,

$$\mathbb{P}\left(\|\hat{\phi}^{\text{Adalina}} - \phi\| \geq \sqrt{\frac{36nD^*C^2}{T} \log \frac{4}{\delta}}\right) \leq \delta.$$

Putting $\sqrt{\frac{36nD^*C^2}{T} \log \frac{4}{\delta}} \leq \epsilon$ yields $T \geq \frac{36nD^*C^2}{\epsilon^2} \log \frac{4}{\delta}$. Setting $\frac{36nD^*C^2}{\epsilon^2} \log \frac{4}{\delta} \geq \frac{16}{9} \log \frac{4}{\delta}$ yields $\epsilon \leq \frac{18C\sqrt{nD^*}}{4}$. Since

$$\sum_{s=1}^{n-1} \sqrt{n\left(\frac{m_s^2}{s} + \frac{m_{s+1}^2}{n-s}\right)} \geq \sum_{s=1}^{n-1} \sqrt{m_s^2 + m_{s+1}^2} \geq \frac{1}{\sqrt{2}} \sum_{s=1}^{n-1} (m_s + m_{s+1}) \geq \frac{1}{\sqrt{2}},$$

we have $D^* \geq \frac{1}{2}$. Therefore, the constraint on $\epsilon$ can be relaxed as $\epsilon \leq 2C$.

$\square$

# B. Bridges

In this appendix, we discuss how our framework bridges the recent approaches (Li & Yu, 2024b; Chen et al., 2025; Fumagalli et al., 2024; Witter et al., 2025).

## B.1. OFA (Li & Yu, 2024b)

The OFA framework makes use of

$$\phi_i = \sum_{s=1}^{n} m_s \cdot \left(\phi_{i,s}^+ - \phi_{i,s-1}^-\right) \quad \text{where} \quad \phi_{i,s}^+ = \mathop{\mathbb{E}}_{\substack{i \in \mathbf{S} \\ |\mathbf{S}|=s}} [U(\mathbf{S})] \quad \text{and} \quad \phi_{i,s-1}^- = \mathop{\mathbb{E}}_{\substack{i \notin \mathbf{S} \\ |\mathbf{S}|=s-1}} [U(\mathbf{S})].$$

Here, each expectations is taken w.r.t. the corresponding uniform distribution. In light of this formula, OFA is proposed to approximate $\{\phi_{i,s}^+, \phi_{i,s}^-\}_{s=1}^{n-1}$ for every $i \in [n]$. It also employs a sampling distribution $\mathbf{q} \in \mathbb{R}^{n-1}$, where $q_s$ denotes the probability of sampling a subset of size $s$ from $2^{[n]}$. Given a sequence of independent sampled subsets $\{S_t\}_{t=1}^T$, OFA proceeds as follows:

$$\hat{\phi}_{i,s}^+ := \frac{\sum_{t=1}^T U(S_t) \cdot [\![i \in S_t, s_t = s]\!]}{T_{i,s}^+} \quad \text{with} \quad T_{i,s}^+ := \sum_{t=1}^T [\![i \in S_t, s_t = s]\!],$$

$$\hat{\phi}_{i,s}^- := \frac{\sum_{t=1}^T U(S_t) \cdot [\![i \notin S_t, s_t = s]\!]}{T_{i,s}^-} \quad \text{with} \quad T_{i,s}^- := \sum_{t=1}^T [\![i \notin S_t, s_t = s]\!].$$

Then,

$$\hat{\phi}_i^{\text{OFA}} := \sum_{s=1}^{n-1} m_s \cdot \hat{\phi}_{i,s}^+ - \sum_{s=1}^{n-1} m_{s+1} \cdot \hat{\phi}_{i,s}^- + (m_n \cdot u_{[n]} - m_1 \cdot u_\emptyset).$$

In particular,

$$\mathbb{E}\left[\frac{T_{i,s}^+}{T}\right] = \frac{s \cdot q_s}{n} \quad \text{and} \quad \mathbb{E}\left[\frac{T_{i,s}^-}{T}\right] = \frac{(n-s) \cdot q_s}{n}.$$

To reduce OFA to a $\Theta(n)$-space version, we first set $\frac{T_{i,s}^+}{T} = \frac{s \cdot q_s}{n}$ and $\frac{T_{i,s}^-}{T} = \frac{(n-s) \cdot q_s}{n}$. Then,

$$\hat{\phi}_{i,s}^+ = \frac{1}{T} \cdot \frac{T}{T_{i,s}^+} \cdot \sum_{t=1}^T U(S_t) \cdot [\![i \in S_t, s_t = s]\!] = \frac{1}{T} \cdot \sum_{t=1}^T \frac{n}{s \cdot q_s} U(S_t) \cdot [\![i \in S_t, s_t = s]\!],$$

$$\hat{\phi}_{i,s}^- = \frac{1}{T} \cdot \frac{T}{T_{i,s}^-} \cdot \sum_{t=1}^T U(S_t) \cdot [\![i \notin S_t, s_t = s]\!] = \frac{1}{T} \cdot \sum_{t=1}^T \frac{n}{(n-s) \cdot q_s} U(S_t) \cdot [\![i \notin S_t, s_t = s]\!].$$

Next, we have

$$\hat{\phi}_i^{\text{OFA}} = \frac{1}{T} \cdot \sum_{t=1}^T U(S_t) \cdot \left(\frac{n \cdot m_{s_t}}{s_t \cdot q_{s_t}} [\![i \in S_t]\!] - \frac{n \cdot m_{s_t+1}}{(n-s_t) \cdot q_{s_t}} [\![i \notin S_t]\!]\right) + (m_n \cdot u_{[n]} - m_1 \cdot u_\emptyset).$$

This exactly recovers our $\hat{\phi}$ in Eq. (3) when Eq. (2) is taken into account.

## B.2. KernelSHAP (Chen et al., 2025)

We demonstrate how their unified formula for unbiased KernelSHAP can be simplified to fit into our framework. Let $\mathbf{A} \in \mathbb{R}^{(2^n-2) \times n}$ be a binary matrix such that $A_{S,i} = 1$ if and only if $i \in S$. Let $\mathbf{M} \in \mathbb{R}^{(2^n-2) \times (2^n-2)}$ be a diagonal matrix such that $M_{S,S} = \frac{n-1}{(n-s)s\binom{n}{s}}$. Additionally, let $\mathbf{Q}$ be any $n \times (n-1)$ matrix such that $\mathbf{Q}^\mathsf{T}\mathbf{Q} = \mathbf{I}$ and $\mathbf{Q}^\mathsf{T}\mathbf{1}_n = \mathbf{0}_n$. Given a

sequence of sampled subsets $\{S_t\}_{t=1}^T$, let $\mathbf{S} \in \mathbb{R}^{T \times (2^n-2)}$ be a sketching matrix such that $S_{t,S_t} = \sqrt{\frac{\binom{n}{s_t}}{T \cdot q_{s_t}}}$ and 0 otherwise.

The unified formula of unbiased kernelSHAP is given as

$$\hat{\phi}^{\text{kernel}} = \mathbf{Q}\mathbf{U}^\mathsf{T}\mathbf{S}^\mathsf{T}\mathbf{S}\mathbf{b}_\lambda + \alpha\mathbf{1}_n$$

$$\text{where } \mathbf{U} := \sqrt{\frac{n}{n-1}}\sqrt{\mathbf{M}}\mathbf{A}\mathbf{Q}, \quad \alpha := \frac{u_{[n]} - u_\emptyset}{n}, \quad \text{and } \mathbf{b}_\lambda := \sqrt{\frac{n}{n-1}}(\sqrt{\mathbf{M}}\mathbf{u} - \lambda\sqrt{\mathbf{M}}\mathbf{A}\mathbf{1}_n).$$

Here, $\lambda \in \mathbb{R}$ is arbitrary. Observe that

$$\mathbf{Q}\mathbf{U}^\mathsf{T}\mathbf{S}^\mathsf{T}\mathbf{S}\mathbf{b}_\lambda = \frac{n}{n-1}\mathbf{Q}\mathbf{Q}^\mathsf{T}\mathbf{A}^\mathsf{T}\sqrt{\mathbf{M}}\mathbf{S}^\mathsf{T}\mathbf{S}\sqrt{\mathbf{M}}(\mathbf{u} - \lambda\mathbf{A}\mathbf{1}_n).$$

For convenience, write $\mathbf{v} = \mathbf{u} - \lambda\mathbf{A}\mathbf{1}_n$. Specifically,

$$\mathbf{Q}\mathbf{Q}^\mathsf{T}\mathbf{A}^\mathsf{T}\sqrt{\mathbf{M}}\mathbf{S}^\mathsf{T}\mathbf{S}\sqrt{\mathbf{M}}\mathbf{v} = \frac{1}{T}\sum_{t=1}^T \frac{\binom{n}{s_t}}{q_{s_t}} \cdot \frac{n-1}{(n-s_t)s_t\binom{n}{s_t}} \cdot v_S \cdot \mathbf{Q}\mathbf{Q}^\mathsf{T}\mathbf{a}_{S_t},$$

where $\mathbf{a}_S$ denotes the $S$-column of $\mathbf{A}^\mathsf{T}$. Since $\mathbf{Q}\mathbf{Q}^\mathsf{T} = \mathbf{I} - \frac{1}{n}\mathbf{J}$, where $\mathbf{J}$ denotes the all-one matrix, we have

$$\left(\mathbf{Q}\mathbf{Q}^\mathsf{T}\mathbf{a}_S\right)_i = \frac{n-s}{n}[\![i \in S]\!] - \frac{s}{n}[\![i \notin S]\!].$$

Therefore,

$$\mathbf{Q}\mathbf{U}^\mathsf{T}\mathbf{S}^\mathsf{T}\mathbf{S}\mathbf{b}_\lambda = \frac{1}{T}\sum_{t=1}^T v_{S_t}\mathbf{z}_{S_t} = \frac{1}{T}\sum_{t=1}^T (u_{S_t} - \lambda s_t)\mathbf{z}_{S_t}.$$

## B.3. SHAP-IQ (Fumagalli et al., 2024)

Given a sequence of sampled subsets $\{S_t\}_{t=1}^T$, the estimate produced by SHAP-IQ is

$$\hat{\phi}_i^{\text{IQ}} := \frac{2H_{n-1}}{T}\sum_{t=1}^T (u_{S_t} - u_\emptyset) \cdot ((n-s_t)m_{s_t}[\![i \in S_t]\!] - s_t m_{s_t+1}[\![i \notin S_t]\!]) + m_n \cdot (u_{[n]} - u_\emptyset),$$

where $H = \sum_{k=1}^{n-1} \frac{1}{k}$. For SHAP-IQ,

$$q_s = \frac{1}{2H_{n-1}} \cdot \frac{n}{s(n-s)}.$$

Then,

$$\hat{\phi}_i^{\text{IQ}} = \frac{1}{T}\sum_{t=1}^T (u_{S_t} - u_\emptyset) \cdot \left(\frac{n}{q_{s_t}} \cdot \frac{m_{s_t}}{s_t}[\![i \in S_t]\!] - \frac{n}{q_{s_t}} \cdot \frac{m_{s_t+1}}{n-s_t}[\![i \notin S_t]\!]\right) + m_n \cdot (u_{[n]} - u_\emptyset).$$

Consequently,

$$\hat{\phi}^{\text{IQ}} = \frac{1}{T}\sum_{t=1}^T (u_{S_t} - u_\emptyset)\mathbf{z}_{S_t} + m_n \cdot (u_{[n]} - u_\emptyset).$$

---

**Algorithm 2** Adalina-All

---

**Input:** Weight vector $\mathbf{m} \in \mathbb{R}^n$ for semi-value $\phi$, total number of samples $T$

**Output:** Estimate $\hat{\phi}^{\text{Adalina}-\text{All}}$

10  Compute the sampling distribution $\mathbf{q} \in \mathbb{R}^{n+1}$ for $0 \leq s \leq n$ such that $q_s \propto \sqrt{\frac{m_s^2}{s} + \frac{m_{s+1}^2}{n-s}}$     // $\frac{x}{0} := 0$

11  Initialize $\hat{\phi}, \hat{\mathbf{v}} \leftarrow \mathbf{0}_n$, $\hat{\gamma} \leftarrow 0$

12  **for** $t = 1, 2, \ldots, T$ **do**

13    Sample a subset size $s$ with probability $q_s$

14    Sample a subset $S$ of size $s$ uniformly from $2^{[n]}$

15    $\hat{\phi} \leftarrow \frac{t-1}{t} \cdot \hat{\phi} + \frac{1}{t} \cdot u_S \mathbf{z}_S^{\text{MSR}}$

16    $\hat{\mathbf{v}} \leftarrow \frac{t-1}{t} \cdot \hat{\mathbf{v}} + \frac{1}{t} \cdot \mathbf{z}_S^{\text{MSR}}$

17    $\hat{\gamma} \leftarrow \frac{t-1}{t} \cdot \hat{\gamma} + \frac{1}{t} \cdot u_S$

18  $\hat{\phi}^{\text{Adalina}-\text{All}} \leftarrow \hat{\phi} - \hat{\gamma}\hat{\mathbf{v}}$

---

## B.4. Regression-Adjusted Approach (Witter et al., 2025)

The sampling distribution $\overline{\mathbf{q}}^{\text{MSR}} \in \mathbb{R}^{n+1}$ used in this approach can be expressed as

$$\overline{q}_s^{\text{MSR}} \binom{n}{s}^{-1} \propto \sqrt{p_{s+1}^2 \left(1 - \frac{s}{n}\right) + p_s^2 \frac{s}{n}} \quad \text{for every } 0 \leq s \leq n.$$

where $p_0$ and $p_{n+1}$ are arbitrary. Observe that

$$p_{s+1}^2 \cdot \left(1 - \frac{s}{n}\right) + p_s^2 \cdot \frac{s}{n} = m_{s+1}^2 \cdot \binom{n-1}{s}^{-1} \binom{n}{s}^{-1} + m_s^2 \cdot \binom{n-1}{s-1}^{-1} \binom{n}{s}^{-1}.$$

Then,

$$\binom{n}{s} \sqrt{p_{s+1}^2 \cdot \left(1 - \frac{s}{n}\right) + p_s^2 \cdot \frac{s}{n}} = \sqrt{n \cdot \left(\frac{m_s^2}{s} + \frac{m_{s+1}^2}{n-s}\right)}.$$

Here, the convention is $\frac{x}{0} := 0$. Eventually, we have

$$\overline{q}_s^{\text{MSR}} \propto \sqrt{\frac{m_s^2}{s} + \frac{m_{s+1}^2}{n-s}} \quad \text{for every } 0 \leq s \leq n.$$

It is clear that $\overline{\mathbf{q}}^{\text{MSR}}$ coincides with $\mathbf{q}^*$, except that $\overline{\mathbf{q}}^{\text{MSR}}$ includes $[n]$ and $\emptyset$ in the sampling pool.

Let

$$D^{\text{MSR}} := \sum_{s=0}^n \frac{n}{\overline{q}_s^{\text{MSR}}} \left(\frac{m_s^2}{s} + \frac{m_{s+1}^2}{n-s}\right).$$

Then,

$$\left(D^{\text{MSR}}\right)^{\frac{1}{2}} = \sum_{s=0}^n \sqrt{n \cdot \left(\frac{m_s^2}{s} + \frac{m_{s+1}^2}{n-s}\right)} = m_1 + (D^*)^{\frac{1}{2}} + m_n,$$

which leads to $(D^*)^{\frac{1}{2}} \leq (D^{\text{MSR}})^{\frac{1}{2}} \leq 2 + (D^*)^{\frac{1}{2}}$. It indicates that $D^{\text{MSR}} \in O(1)$ if and only if $D^* \in O(1)$. Therefore, according to Li & Yu (2024b, Proposition 4), $D^{\text{MSR}} \in O(1)$ for Beta Shapley values and weighted Banzhaf values.

Under the use of $\overline{\mathbf{q}}^{\text{MSR}}$, $\mathbf{z}_S^{\text{MSR}} = \mathbf{z}_S$ with $q_s = \overline{q}_s^{\text{MSR}}$ for all $\emptyset \subsetneq S \subsetneq [n]$, while $\mathbf{z}_\emptyset^{\text{MSR}} = -\frac{m_1}{\overline{q}_0^{\text{MSR}}}\mathbf{1}_n$ and $\mathbf{z}_{[n]}^{\text{MSR}} = \frac{m_n}{\overline{q}_n^{\text{MSR}}}\mathbf{1}_n$. In this case, one can verify that $\|\mathbf{z}_S\|_2^2 = nD^{\text{MSR}}$ for every $S \subseteq [n]$. Moreover,

$$\mathbb{E}[\mathbf{z}_S^{\text{MSR}}] = \mathbf{0}_n$$

*Table 1.* Summary of the datasets used.

| Dataset | #Instances | #Features | Source | Task | #Classes | Depth |
|---|---|---|---|---|---|---|
| FOTP | 6,118 | 51 | https://openml.org/d/1475 | classification | 6 | 20 |
| GPSP | 9,873 | 32 | https://openml.org/d/4538 | classification | 5 | 10 |
| MinibooNE | 130,064 | 50 | https://openml.org/d/41150 | classification | 2 | 20 |
| philippine | 5,832 | 308 | https://openml.org/d/41145 | classification | 2 | 15 |
| spambase | 4,601 | 57 | https://openml.org/d/44 | classification | 2 | 15 |
| jannis | 83,733 | 54 | https://openml.org/d/41168 | classification | 4 | 40 |
| har | 10,299 | 561 | https://openml.org/d/1478 | classification | 6 | 20 |
| Fashion-MNIST | 70,000 | 784 | https://openml.org/d/40996 | classification | 10 | 30 |
| CIFAR10_small | 20,000 | 3,072 | https://openml.org/d/40926 | classification | 10 | 20 |
| superconduct | 21,263 | 81 | https://openml.org/d/43174 | regression | - | 10 |
| wave_energy | 72,000 | 48 | https://openml.org/d/44975 | regression | - | 30 |
| BT | 583,250 | 77 | https://openml.org/d/4549 | regression | - | 20 |

for every semi-value, which is one of the keys to prove Theorem 4.1. As a result, Theorems 3.1 and 4.1 continue to hold when using $\overline{\mathbf{q}}^{\mathrm{MSR}}$, where $D^*$ and $\|\boldsymbol{\varphi}\|_2^2$ are replaced by $D^{\mathrm{MSR}}$ and $\|\boldsymbol{\phi}\|_2^2$, respectively, and the constraint on symmetric semi-values is removed. After all, the corresponding estimate is

$$\hat{\boldsymbol{\phi}}^{\mathrm{MSR}} := \frac{1}{T} \sum_{t=1}^{T} u_{S_t} \mathbf{z}_{S_t}^{\mathrm{MSR}}.$$

Its adaptive version is given by

$$\hat{\boldsymbol{\phi}}^{\mathrm{MSR-adaptive}} := \frac{1}{T} \sum_{t=1}^{T} (u_{S_t} - \hat{\gamma}) \cdot \mathbf{z}_{S_t}^{\mathrm{MSR}} \quad \text{where} \quad \hat{\gamma} := \frac{1}{T} \sum_{t=1}^{T} u_{S_t}.$$

The corresponding procedure is presented in Algorithm 2.

## C. Experiments

For kernelSHAP, we set $\mathbf{q} = \mathbf{q}^*$ and $\lambda = \frac{u_{[n]} - u_\emptyset}{n}$, a combination that is empirically the best according to Chen et al. (2025). For the results in Figure 1, the number of trees is set to 50 when utility functions $U$ may take both positive and negative values; to ensure that $U(S) > 0$ for every $S$, we instead employ `DecisionTreeClassifier` from the scikit-learn library. The details of the employed datasets are summarized in Table 1, which includes the depth of trees used to construct utility functions. Additional results on the comparison of randomized algorithms for approximating semi-values are shown in Figures 5, 6 and 7.

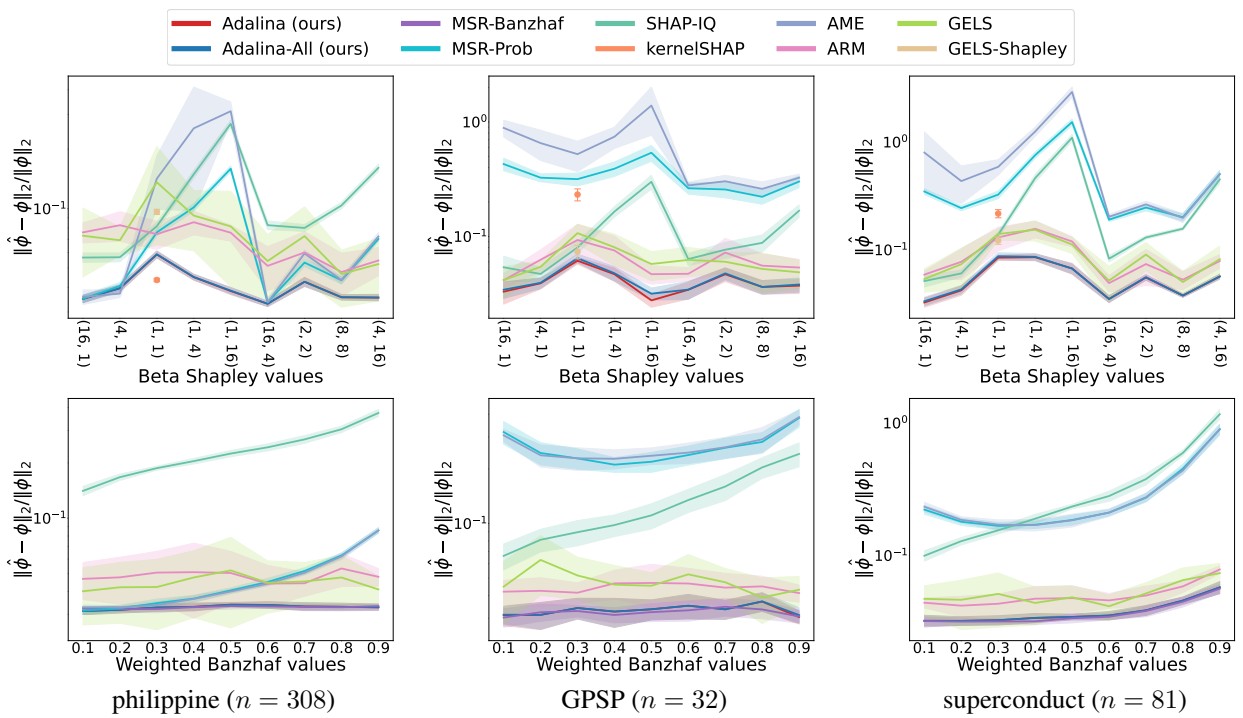

*Figure 5.* The relative approximation error of different randomized algorithms. Weighted Banzhaf values are parameterized by $w \in (0, 1)$, whereas Beta Shapley values are parameterized by $(\alpha, \beta)$, with $(1, 1)$ corresponding to the Shapley value.

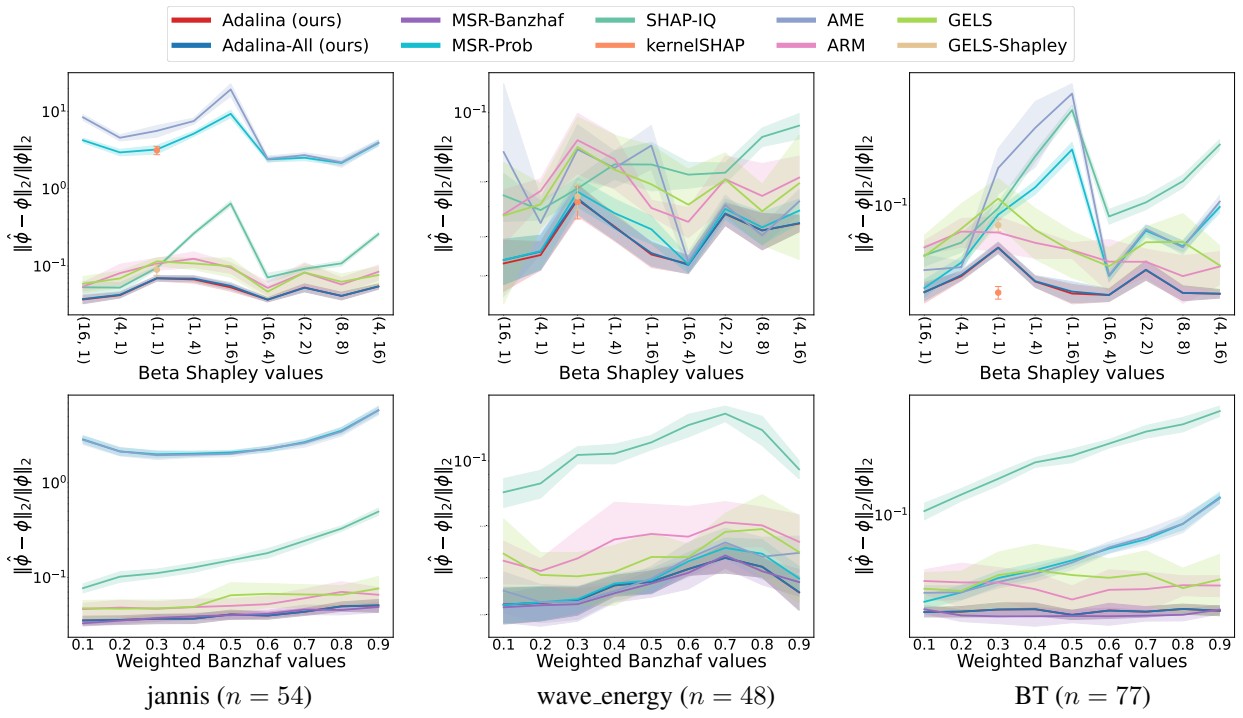

*Figure 6.* The relative approximation error of different randomized algorithms. Weighted Banzhaf values are parameterized by $w \in (0, 1)$, whereas Beta Shapley values are parameterized by $(\alpha, \beta)$, with $(1, 1)$ corresponding to the Shapley value.

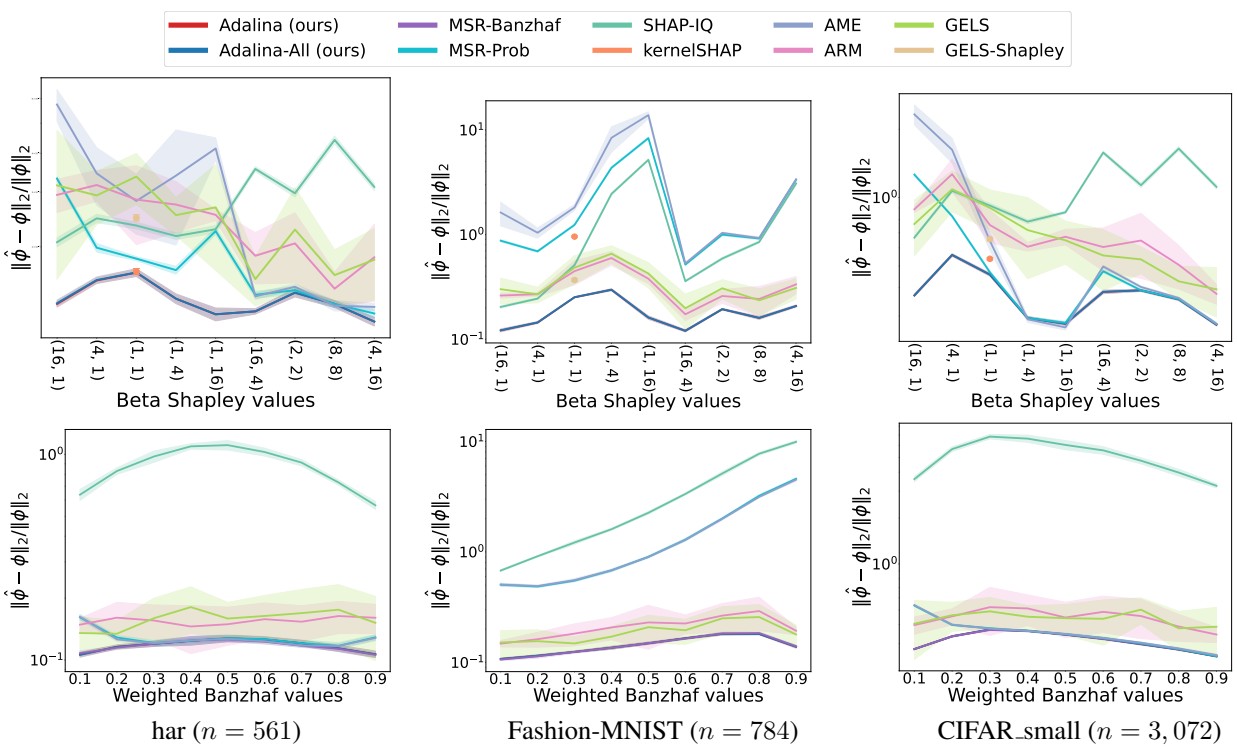

*Figure 7.* The relative approximation error of different randomized algorithms. Weighted Banzhaf values are parameterized by $w \in (0, 1)$, whereas Beta Shapley values are parameterized by $(\alpha, \beta)$, with $(1, 1)$ corresponding to the Shapley value.

