# OpenReview forum: "Adalina: Adaptive Linear Approximation for the Shapley Value and Beyond"
_ICML.cc/2026/Conference — ICML 2026 regular_

### Official Review · Reviewer_81zp · 2026-03-09

**Soundness:** 3
**Presentation:** 2
**Significance:** 2
**Originality:** 2
**Overall Recommendation:** 2
**Confidence:** 5

**Summary:**

Setup: The paper considers the problem of approximating Shapley values. Let $U:{0,1}^n to \mathbb{R}$ be a boolean function with Shapley values $\phi_i(U)$ for $i \in [n]$. The paper considers algorithms to design estimates $\hat{\phi}_i(U)$ in linear space in $n$ that a) have low expected squared error and b) close to the true Shapley values with high probability.

First, they consider a vector concentration inequality, rather than the matrix or scalar concentration inequalities used in prior work. Using this inequality, they revisit the analysis of several prior Shapley value estimators that are in the vein of "MSR" or "unbiased kernelSHAP". That is, these estimators estimate the sum:
$$
\sum_S U(S) (1[i \in S] p_|S| - 1[i \notin S] p_{|S|-1}.
$$
Using this tighter analysis, they get guarantees that $O(n/epsilon^2 \log(1/\delta))$ queries are needed rather than $O(n/epsilon^2 \log(n/\delta))$ to get an $\epsilon, \delta$ guarantee. That is, they shave off a logarithmic factor in $n$.

Second, they propose an estimator called Adalina that is a variant of the MSR/unbiased kernelSHAP estimator. It subtracts off a value gamma from the the estimates to reduce their variance, then adds it back so that the final estimator is unbiased. Instead of fixing gamma, the work adaptively chooses gamma as the average of the $u$ values.

In addition, they analyze the effect of paired sampling on the variance of their estimators. They show that paired sampling estimates the Shapley values of:
$$
U(S) = \frac{U(S)-U(S^c)}{2}
$$
where $S^c = [n] \setminus S$ is the complement of $S$.

**Compliance With Llm Reviewing Policy:**

Affirmed.

**Final Justification:**

While I appreciate the engagement from the authors, I ultimately recommend rejection because:

1. They did not adequately compare to Regression MSR of Witter et al. 2025. When they did at the end of the rebuttal session, they found it was better than their algorithm.

2. Their claims of unifying prior MSR algorithms and improving the analysis are simple and straightforward in my opinion.

3. Their whole goal is to subtract the mean from the variance. While they estimate this mean to subtract in their algorithm, you can accomplish the same result using exactly $T$ without replacement samples with a much simpler estimator. There is a little work in generalizing this strategy (i.e., scalars to vectors and a different independent sum for each subset size), but it is quite straightforward.

**Key Questions For Authors:**

1. How is your estimator unbiased if you're not adding back $\gamma$?

2. Do you see how *not* subtracting and adding back the constant $\gamma$ but using exactly $T$ samples results in variance where the mean is automatically being subtracted?

3. Could you please double check your implementation of Regression MSR to make sure it's aligned with what Witter et al. 2025 describe?

**Limitations:**

Yes.

**Strengths And Weaknesses:**

Here are my thoughts in chronological (to the way my brain works) order:

1. **Connection to literature:** I *really* appreciate how closely this paper engages with prior work: It's clear the authors have carefully read the relevant literature especially Chen et al. 2025 paper, Witter et al. 2025, Lin et al. 2022, etc.

2. **Vector Concentration:** I think tighter analysis is generally better. In the paper, they get a tighter analysis of prior works by using a standard vector concentration inequality instead of standard scalar inequalities. The result is removing logarithmic factors from the query guarantees. For any $n$, especially for the $n \leq 1000$ used in explainable AI, log n is basically a constant so I don't think is much of a contribution, especially at a premier ML conference like ICML.

3. **Paired Sampling:** I appreciate Theorem 3.2 on paired sampling. In particular, the statement they get (by using a more careful analysis) is that unbiased kernelSHAP estimators under paired sampling are effectively estimating the *odd* components of the Boolean function $U$. This is very closely related to the contemporary work "An Odd Estimator for Shapley Values". There, the authors show that the Shapley values of $U$ are simply the Shapley values of the odd component of $U$. They combine this idea with surrogate models to estimate only the odd component of $U$, which (as you show too) is exactly what paired sampling is doing.

4. **Mean Adjustment:** The key component of Adalina is to subtract off $\gamma$ from their estimates i.e.,
$$
\hat{\mathbf{\phi}} =\gamma + \left( \frac1{T} \sum_{t=1}^T (U(S_t) -\gamma) \mathbf{z}_{S_t} \right)+ m_n(U([n]) - \gamma) - m_0 (U(\emptyset) - \gamma).
$$

**Note:** I think the paper doesn't include the initial $\gamma$, which would make the estimator biased. One way to see this: in the paper's formulation without the initial $\gamma$, we could set $\gamma$ to be an arbitrarily large constant which would clearly bias the estimator. As I wrote it, the estimator is correct in expectation. Anyway, the point is that you can reduce the variance of the estimator if $\gamma$ is chosen well. For constant $\gamma$, the best you can do is the mean. So the authors estimate the mean in the algorithm.

**Very Important:** If you always sample exactly $T$ values, the variance of the standard estimator (without subtracting $\gamma$) actually depends on $\mathbb{E}[(U(S)-\bar{U})^2]$ where $\bar{U}$ is the mean (wrt the distribution of the expectation/sampling) of $U$. It's easiest to see this on a simple mean estimation problem where you can write the standard estimate as subtracting the mean $\bar{U}$ in the summation and then adding it back afterward. This is particularly cool because you don't need to know $\bar{U}$ for this to work, all you need is that you're always taking exactly $T$ samples. Since their $\gamma$ is always an estimate of $\bar{U}$, the standard estimator with exactly $T$ samples should always have lower variance.

In fact, the "next step" from realizing you can reduce the variance by subtracting (and adding back) a constant is to subtract and add back a surrogate function. This is exactly what the Regression MSR strategy in Witter et al. 2025 does, which brings me to:

5. **Implementation of Regression MSR:** I'm comparing your Figure 4 to Figure 3 in Witter et al. 2025. In your Figure 4, what you call MSR-Witter is generally better than SHAP-IQ, comparable to MSR-Wang, and worse than AME, ARM, and GELS. In the Figure 3 of Witter et al. 2025, TreeMSR (I assume what you're calling MSR-Witter?) is better than all the above by several orders of magnitude. I'm concerned that you're not implementing this algorithm correctly. In particular, what you call $V$ in line 313 should be a gradient boosted tree learned on the samples of $U$ and $\phi(V)$ should be the Shapley values of this GBT.

---

> ### Author Rebuttal · Authors · 2026-03-31
>
> We thank Reviewer 81zp for the careful examination.
>
> Q1: How is your estimator unbiased if you're not adding back $\gamma$?
> - There is some confusion here: we do not add $\gamma$ back since
> $$E[\frac{1}{T}\sum_t(u(S_t)-\gamma)z_{S_t}]=\varphi,$$ where we ued the fact that $E[z_{S_t}] = 0$. Alternatively, following Witter et al., what we should add back is $\phi(\gamma1)\equiv0$, i.e., by setting $V(S)\equiv\gamma$.
> - Our estimator $\hat\phi^\gamma$ is unbiased if $\gamma$ is deterministic. For Adalina, $\hat\gamma$ correlates with $S_t$, becoming biased. Nevertheless, Adalina is asymptotically unbiased, when $T\to\infty$.
>
> Q2 : Do you see how not subtracting and adding back the constant $\gamma$ but using exactly $T$ samples results in variance where the mean is automatically being subtracted?
> - We are not sure if we understand this question correctly. We remind again that we cannot add $\gamma$ back to our estimator
> $$\hat\phi^\gamma=\frac{1}{T}\sum_t(u_{S_t}-\gamma)z_{S_t} + b,$$ where we wish to emphasize the **random vector** $z_{S_t}$. The variance of $\hat\phi^\gamma$ (for deterministic $\gamma$) is
> $\frac{E[ (u_{S_t}-\gamma)^2\cdot\\|z_{S_t}\\|_2^2]-\\|\varphi\\|_2^2}{T}$, so the optimal $\gamma^\*=\frac{E[u\_{S_t}\cdot\\|z\_{S_t}\\|_2^2]}{E[\\|z\_{S_t}\\|_2^2]}$, which is not necessarily $E[u\_{S_t}]$. Only with the optimal sampling distribution $q^\*$, $\\|z\_{S_t}\\|$ is constant (see Theorem 3.1) and then $\gamma^\*$ reduces to $E[u\_{S_t}]$. This highlights the importance of using $q^*$.
> - In Theorem 3.1, without substracting $\gamma^\*$, $E[\\|\hat{\phi} - \phi\\|_2^2]=\frac{nD^\* E[u_S^2]-\\|\varphi\\|_2^2}{T}$. By contrast, after substracting $\gamma^*$, it becomes $E[\\|\hat{\phi}^{\gamma^\*}-\phi\|_2^2]=\frac{nD^\* E[(u_S-\gamma^\*)^2]-\\|\varphi\\|_2^2}{T}$ instead. These two quantities are not equal, and we cannot achieve smaller mean square error automatically.
>
> Q3: Could you please double check your implementation of Regression MSR to make sure it's aligned with what Witter et al. 2025 describe? I assume what you're calling MSR-Witter is TreeMSR.
> - MSR-Witter is not TreeMSR. Our MSR-Witter corresponds to Eq. (3) in (Witter et al.) where $\mathcal{D}(S)$ is specified on their page 6 (arxiv). In particular, MSR-Witter does not involve any regression. The details of TreeMSR will be described below.
> - Witter et al. proposed using a class of simple utility functions (whose semi-values can be computed) to approximate $U$ as $V$. Since $\phi(U)=\phi(V)+\phi(U-V)$, the estimate $\hat\phi(U)$ can be computed as $\hat\phi(U)=\phi(V)+\hat\phi(U-V)$. When $V$ is restricted to decision trees and the estimate $\hat\phi(U-V)$ is yielded using **MSR-Witter**, this procedure is referred to as **treeMSR**. In particular, the estimator for approximating $\phi(U-V)$ is replaceable, and the estimation error reduces to $\phi(U)-\hat\phi(U)=\phi(U-V)-\hat\phi(U-V)$.
>
> Q4: In fact, the "next step" from realizing you can reduce the variance by subtracting (and adding back) a constant is to subtract and add back a surrogate function. This is exactly what the Regression MSR strategy in Witter et al. does.
> - Correct. Adalina fits the regression approach of Witter et al., where we replace their tree with a constant function. However, Adalina brings several advantages:
>   - choosing a constant function allows us to give a simple, **adaptive**  estimator, where regression and estimation are done simultaneously in a single pass. In contrast, TreeMSR splits the sampled subsets to 2 stages, one for regression and the other for estimating the value of the residual utility.
>   - due to its simplicity, we are able to analyze Adalina rigorously, see our Theorem 4.1 which showed how much Adalina improves the mean square error. We believe this is a nontrivial improvement over Theorem 2.1 of Witter et al.
>   - Adalina could also be used to improve Witter et al: simply apply Adalina to estimate the value of their residual utility, possibly leading to double variance reduction.
>
> Q5: For any $n$, especially for the  $n\leq 1000$ used in explainable AI, $\log n$ is basically a constant so I don't think is much of a contribution.
> - This is an improvement in analysis technique that we hope to draw attention to the community as it applies to many existing algorithms; it induces no change in the actual implementation of algorithms and hence is kind of a free lunch.
> - Even for small $n$ such as $1000$, $\ln(1000)\approx 7$, so we can ascertain a 7x reduction in the number of utility evaluations, which we argue is a highly nontrivial improvement for practitioners. Not to mention that besides explainable AI, semi-values has also been empirically studied in, e.g., dataset refinement (He et al., 2024) or text attribution (CohenWang et al., 2024) where $n$ can be much larger.
> - More importantly, removing the log-factor allows us to discern the optimality of existing and newly proposed algorithms.

---

> > ### Author Rebuttal · Reviewer_81zp · 2026-04-01
> >
> > I thank the authors for their response. There are several points that are unresolved:
> >
> > ### Baseline Comparison Name
> >
> > The authors implement the MSR estimator, and call this "MSR-Witter" because it appears in Equation 3 of Witter et al. 2025. If you read the surrounding text, you'll realize that this is an introduction describing prior work on estimating Shapley values. In particular, the MSR algorithm was proposed for Banzhaf values by Wang and Jia 2023 and extended to other probabilistic values by a line of more recent work.
> >
> > **Calling this estimator MSR-Witter is misleading and wrong.** First, you are falsely attributing the estimator to a paper that simply (and quite clearly) described the estimator and where it came from. Second, you are falsely making it appear *as if* you implemented the actual algorithm proposed by Witter et al. 2025.
> >
> > **I would like to believe the authors did this unintentionally, but I am very concerned that they did not correct the mistake in their response to my review.**
> >
> > ### Log Factors in the Analysis
> >
> > The authors defend their contribution of the log factor improvement by saying that $\ln(1000) \approx 7$ and it would be nice if the algorithms improved by this factor in experiments. To make this claim, they would need to actually compare to SOTA Shapley value estimators.
> >
> > In addition, I really don't think the log factor improvement is notable when a) you're already hiding constants in your theoretical statements and b) the more careful analysis is simply a result of using a more specific theorem from prior work.
> >
> > ### Comparison to Regression MSR
> >
> > The authors acknowledge that the Regression MSR (the actual algorithm proposed by Witter et al. 2025) is a kind of generalization of theirs. Namely, instead of subtracting a constant of the mean to reduce variance, Regression MSR subtracts a learned approximation to the values. In addition, the Regression MSR is SOTA (or close to SOTA) for Shapley value estimation see, e.g., recent preprints like "PolySHAP: Extending KernelSHAP with Interaction-Informed Polynomial Regression" and "An Odd Estimator for Shapley Values".
> >
> > **For both reasons, the authors should implement this algorithm and compare to it.**
> >
> > ### **Automatic Variance Centering**
> >
> > I thank the authors for correcting my misunderstanding about the bias of their estimator.
> >
> > My larger point is that I think their variance centering can be better achieved with a more simple estimator: As I understand it, their algorithm is fundamentally a Monte Carlo estimator where the *variance is reduced by an estimate of the mean*. I believe that a standard stratified estimator with exactly $T$ samples *sampled without replacement* achieves *variance reduced by the true mean*. I did not explain the idea behind this well in my initial review, and I will endeavor to correct that now.
> >
> > Consider the standard problem of mean estimation: there are $N$ values $x_1, \ldots, x_N$ and we would like to estimate the true mean $\mu= \frac1{N} \sum_{i=1}^N x_i$. We are given a budget of $T$ samples and we construct the estimator:
> >
> > $$
> > \hat{\mu} = \frac1{T} \sum_{i=1}^N I_i x_i
> > $$
> >
> > where $I_i$ is the indicator that the $i$th value is sampled. If exactly $T$ samples are drawn without replacement, then the variance of the estimator is
> >
> > $$
> > \text{Var}(\hat{\mu}) = \frac{N-T}{T \cdot N(N-1)} \sum_{i=1}^N \Big( x_i - \mu \Big)^2.
> > $$
> >
> > This is a kind of magical result, and I highly recommend the authors prove this for themselves. The incredible thing is that, without knowing $\mu$, the *variance* of the estimator gets reduced by $\mu$. Fundamentally, this happens because the covariance between the sampling indicators is actually negative and leads to this elegant cancellation." (Specifically, $\text{Cov}(I_i, I_j) = \frac{T(T-1)}{N(N-1)} - \frac{T^2}{N^2} < 0$.)
> >
> > Now, the problem of estimating Shapley values is not *exactly* mean estimation. But the generalization is somewhat straightforward: Use vectors instead of the scalars $x_i$, and view the Shapley value estimator as a *stratified* sampler i.e., each subset size is sampled and estimated separately.
> >
> > Given the very close connection to their algorithm, I think the authors should compare to this straightforward estimator both in a) the theoretical statement of the variance and b) the experiments.

---

> > > ### Author Response · Authors · 2026-04-06
> > >
> > > We thank the reviewer for further clarification. We will incorporate all discussions (including other comments ommitted due to space) in our revision.
> > >
> > > Q1: About MSR-Witter
> > > - We agree this name can be misleading and will change it to MSR-General. Thank you for pointing it out. We wish to note that since we used a tree-based utility in larger-scale experiments (in order to compare to groundtruth), treeMSR of Witter et al. has a significant advantage (almost perfectly fitting the tree-utility, resulting in almost exact solution).
> > > - We conduct additional experiment to support our claim that our Adalina-W, which is an adaptive version of MSR-General, can be used to enhance treeMSR. We strictly follow the procedure of Witter et al. (Algorithm 1). Empirically, trees easily fit tree-based utility. As a result, the residual $\phi(U-V)$ can not distinguish different estimators. So, we use the SOU utility from Li and Yu (2024b) instead.
> > >
> > > n=50, T=40n||||||||||
> > > -|-|-|-|-|-|-|-|-|-
> > > **Estimator**|**(16,1)**|**(4,1)**|**(1,1)**|**(1,4)**|**(1,16)**|**(16,4)**|**(2,2)**|**(8,8)**|**(4,16)**
> > > fitted $V$ + MSR-General (treeMSR) |0.00|0.01|0.18|0.29|0.43|0.01|0.10|0.02|0.19
> > > fitted $V$ + Adalina-W |0.00|0.01|0.10|0.12|0.35|0.01|0.10|0.02|0.09
> > > Adalina|0.00|0.02|0.24|0.35|0.47|0.01|0.10|0.02|0.15
> > > **n=100**
> > > treeMSR|0.00|0.01|0.20|0.29|0.60|0.00|0.07|0.01|0.10
> > > fitted $V$ + Adalina-W|0.00|0.01|0.12|0.14|0.49|0.00|0.07|0.01|0.04
> > > Adalina|0.00|0.01|0.21|0.36|0.57|0.00|0.07|0.01|0.08
> > >
> > >
> > > Q2: Comparison to Regression-MSA
> > > - As stated in our contributions, our framework not only bridges regression-MSR, but also (i) connects to OFA, SHAP-IQ and the unified unbiased kernelSHAP, (ii) provides insights on when paired sampling is beneficial, and (iii) theoretically establishes improved complexity. **We wish to emphasize the key differences from Regression-MSR: (a) Adalina is online (one-pass, never stores any subset) while Regression-MSR splits sampled subsets into two stages; (b) Adalina is strictly a linear space (with no dependence on $\epsilon$ or $\delta$) and linear time algorithm; \(c) Adalina applied control variates to reduce variance (see Q2 to Reviewer 8CR1); (d) Adalina has stronger theoretical guarantee.**
> > >
> > > Q3: Automatic variance centering
> > >
> > > Thank you for suggesting sampling w/o replacement. We wish to point out the following:
> > > - Sampling w/o replacement does not achieve the same variance reduction as ours (control variates). In the reviewer's example, w/o replacement decreases the variance by a multiplicative factor $\frac{N-T}{N-1}$, but there is no effect of "automatic variance centering" by the true mean.
> > > - Sampling w/o replacement yields provably smaller variance when sampling **uniformly** (Hoeffding 1963, $\S 6$), as in the reviewer's example. In Adalina, we sample the subsets *non-uniformly*, making w/o replacement costly to implement, especially if we have  space or time constraints.
> > > - We conducted experiment to show Adalina is amenable to sampling w/o replacement. *This is another evidence that the latter cannot fully explain the success of Adalina*. In the table below, we compare
> > >   - baseline, by rewriting $\varphi$ as a *uniform* average over $x_S=2^nw_S u_S$, and use the suggested w/o replacement scheme. Note, however, that each $x_S$ is magnified by $2^n$ , potentially leading to a much larger variance.
> > >   - w/o ($\lambda=0$), by following previous  w/o replacement scheme  (Musco & Witter, 2025; Chen et al., 2025). Specifically,$$\hat\varphi = \sum_S Y_S\frac{w_S (u_S-\lambda)}{b_S},$$ where $\{Y_S\}$ are independent Bernoulli with $P(Y_S=1)= b_S := \min(1,c\cdot q_S\binom{n}{s}^{-1})$ and $c$ is chosen so that $\sum_S b_S = T$, the number of distinct subsets.
> > >   - adaptive w\o, by following Chen et al. (2025, Theorem A.11) and assuming $q^\*$ is used (Eq. (5)), we can show $$Var(\hat\varphi) \leq \frac{nD^\*\sum_Sq_S^*\binom{n}{s}^{-1}(u_S-\lambda)^2}{T},$$ leading to the same $\lambda^\*$ as in Adalina. The practical difference is that $\lambda^\*$ is estimated as $\sum_S Y_S\frac{q^\*_S \binom{n}{s}^{-1} u_S}{b_S}$.
> > >   - Adalina: sampling w/ replacement
> > >
> > > Since sampling w/o replacement does not scale, we use smaller datasets from openml. Adaptive w/o seems to perform the best, followed by Adalina.
> > >
> > > wall-robot-navigation, n=24, T=1000n ||||||||||
> > > -|-|-|-|-|-|-|-|-|-
> > > **Estimator**|**(16,1)**|**(4,1)**|**(1,1)**|**(1,4)**|**(1,16)**|**(16,4)**|**(2,2)**|**(8,8)**|**(4,16)**
> > > baseline|2432.170|877.567|34.821|990.326|2741.315|204.840|7.952|0.242|232.502
> > > w\o|0.018|0.078|0.094|0.074|0.018|0.095|0.112|0.113|0.093
> > > adaptive w\o|0.001|0.004|0.004|0.002|0.000|0.005|0.005|0.005|0.002
> > > Adalina|0.010|0.009|0.007|0.003|0.001|0.009|0.005|0.005|0.002
> > > **letter, n=16**|
> > > baseline|6.096|2.492|0.594|2.974|7.400|1.029|0.226|0.066|1.157
> > > w\o|0.001|0.017|0.029|0.016|0.001|0.012|0.038|0.050|0.010
> > > adaptive w\o|0.000|0.001|0.001|0.001|0.000|0.001|0.002|0.002|0.001
> > > Adalina|0.007|0.006|0.005|0.002|0.001|0.005|0.004|0.003|0.002

---

### Official Review · Reviewer_AAnM · 2026-03-12

**Soundness:** 3
**Presentation:** 2
**Significance:** 3
**Originality:** 3
**Overall Recommendation:** 5
**Confidence:** 2

**Summary:**

This paper studies the problem of approximating Shapley Values and semi values. The authors introduce a new theoretical framework based on a vector concentration inequality which improves the asymptotic query complexity of existing algorithms to $\mathcal{O}(\frac{n}{\epsilon^2}\log(\frac{1}{\delta}))$ where the memory constraint is $\mathcal{O}(n)$.

Using this framework, they propose a randomized algorithm called Adalina for approximating semi values. The algorithm retains the same theoretical query complexity and achieves better empirical results.  Finally, the paper evaluates Adalina against several existing approximation techniques.

**Compliance With Llm Reviewing Policy:**

Affirmed.

**Final Justification:**

The authors proposed a new framework to analyze linear approximations of Shapley values. This work reduces asymptotic query complexity in approximating Shapley values by logarithmic factors.

In their rebuttal, the authors have run additional experiments to show that their method scales. I am satisfied with their response and will maintain my score.

**Key Questions For Authors:**

1) In the experiments, the largest number of features is 308. Have you tried your algorithm on larger datasets with more features? How well would it scale?

**Limitations:**

yes

**Strengths And Weaknesses:**

- The paper provides a strong theoretic contribution by improving the asymptotic query complexity in approximating Shapley values to $\mathcal{O}(\frac{n}{\epsilon^2}\log(\frac{1}{\delta}))$ under linear space constraints. That is memory scales as $\mathcal{O}(n)$ with number of players.

- The proposed algorithm Adalina also achieves the same asymptotic query complexity of $\mathcal{O}(\frac{n}{\epsilon^2}\log(\frac{1}{\delta}))$ while also improving upon the approximation variance.

- The paper provides empirical evaluations comparing the proposed algorithm with several existing baselines for approximating Shapley values and other semi values. The results show that the proposed method achieves the lowest approximation error in most cases, although for the regular Shapley value unbiased kernelSHAP can perform better.

- Some of the inline equations are a bit hard to read. For instance, the equations
   at the bottom of page 3 and 4.

- The experiments are on a smaller scale. The largest number of features was 308. It would be helpful to see the performance of the method in larger scale problems.

- A section highlighting the computational overhead of the various methods empirically would be helpful.

---

> ### Author Rebuttal · Authors · 2026-03-31
>
> We thank Reviewer AAnM for the helpful suggestion. We will improve the readability of equations in the revision.
>
> Q1: The experiments are on a smaller scale. The largest number of features was 308. It would be helpful to see the performance of the method in larger scale problems. In the experiments, the largest number of features is 308. Have you tried your algorithm on larger datasets with more features?
>
> We employ three additional datasets from openml: (i) har (n=561), (ii) Fashion-MNIST (n=783), and (iii) CIFAR_10_small (n=3,072).
>
> **CIFAR_10_small, n=3,072**|||||||||||||||||||
> -|-|-|-|-|-|-|-|-|-|-|-|-|-|-|-|-|-|-|
> **Estimator**|**(16,1)**|**(4,1)**|**(1,1)**|**(1,4)**|**(1,16)**|**(16,4)**|**(2,2)**|**(8,8)**|**(4,16)**|**0.1**|**0.2**|**0.3**|**0.4**|**0.5**|**0.6**|**0.7**|**0.8**|**0.9**
> Adalina|0.24|0.40|0.31|0.22|0.20|0.26|0.29|0.28|0.21|0.19|0.23|0.25|0.26|0.26|0.24|0.22|0.20|0.18
> MSR-Wang||||||||||0.19|0.23|0.25|0.27|0.26|0.24|0.22|0.20|0.18
> MSR-Witter|2.05|1.32|0.44|0.25|0.21|1.02|0.46|0.50|0.25|1.15|0.97|0.83|0.66|0.52|0.41|0.32|0.26|0.20
> SHAP-IQ|0.41|0.58|0.43|0.40|0.43|0.82|0.57|0.88|0.59|2.01|2.92|3.61|3.36|3.13|3.08|2.68|2.16|1.58
> kernelSHAP|||0.56
> AME|3.37|2.01|0.52|0.24|0.19|1.04|0.46|0.51|0.25|1.15|0.98|0.84|0.68|0.53|0.41|0.32|0.26|0.20
> ARM|0.58|0.80|0.48|0.46|0.47|0.36|0.41|0.44|0.34|0.26|0.28|0.37|0.44|0.29|0.32|0.29|0.30|0.27
> GELS|0.52|0.93|0.68|0.51|0.57|0.39|0.32|0.30|0.25|0.25|0.26|0.29|0.43|0.31|0.27|0.31|0.28|0.29
> GELS-Shapley|||0.52
> **Fashion-MNIST, n=783**
> Adalina|0.06|0.07|0.10|0.12|0.17|0.06|0.08|0.07|0.09|0.05|0.06|0.06|0.06|0.06|0.07|0.07|0.08|0.10
> MSR-Wang||||||||||0.05|0.06|0.06|0.06|0.06|0.07|0.07|0.08|0.10
> MSR-Witter|0.26|0.16|0.14|0.12|0.18|0.12|0.10|0.09|0.09|0.13|0.11|0.10|0.09|0.09|0.08|0.08|0.08|0.10
> SHAP-IQ|0.10|0.11|0.19|0.60|2.67|0.16|0.20|0.27|0.61|0.34|0.41|0.51|0.59|0.69|0.84|1.02|1.42|2.77
> kernelSHAP|||0.21
> AME|0.47|0.27|0.22|0.10|0.13|0.12|0.11|0.09|0.09|0.14|0.11|0.10|0.09|0.09|0.08|0.08|0.08|0.10
> ARM|0.13|0.14|0.14|0.26|0.41|0.08|0.10|0.08|0.13|0.08|0.08|0.07|0.07|0.08|0.09|0.09|0.10|0.13
> GELS|0.16|0.12|0.18|0.31|0.35|0.08|0.13|0.08|0.14|0.08|0.08|0.07|0.08|0.08|0.08|0.09|0.10|0.16
> GELS-Shapley|||0.16
> **har, n=561**
> Adalina|0.07|0.09|0.11|0.07|0.06|0.06|0.08|0.06|0.06|0.05|0.05|0.06|0.05|0.06|0.05|0.05|0.05|0.05
> MSR-Wang||||||||||0.05|0.05|0.05|0.05|0.05|0.05|0.05|0.05|0.05
> MSR-Witter|0.24|0.16|0.19|0.21|0.31|0.13|0.14|0.11|0.14|0.14|0.12|0.11|0.11|0.11|0.11|0.11|0.13|0.16
> SHAP-IQ|0.10|0.12|0.17|0.26|0.44|0.17|0.17|0.22|0.29|0.30|0.37|0.45|0.49|0.53|0.54|0.61|0.62|0.72
> kernelSHAP|||0.07
> AME|0.38|0.48|0.30|0.46|0.60|0.12|0.15|0.11|0.14|0.14|0.12|0.11|0.11|0.11|0.11|0.11|0.13|0.17
> ARM|0.16|0.25|0.21|0.14|0.13|0.09|0.10|0.08|0.08|0.07|0.07|0.08|0.08|0.09|0.07|0.07|0.08|0.07
> GELS|0.11|0.20|0.27|0.16|0.11|0.09|0.13|0.09|0.07|0.08|0.09|0.08|0.07|0.07|0.07|0.07|0.08|0.07
> GELS-Shapley|||0.19
>
> Q2: How well would it scale?
> - We have maken it explicitly in our revision that, for any $0<\epsilon\leq2C$ (assume $\\|U\\|_\infty\leq C$) and any $0<\delta<1$, Adalina-W requires $\frac{16nD^* C^2}{\epsilon^2}\log\frac{4}{\delta}$ utility evaluations to achieve $P(\\|\hat\phi-\phi\\|_2^2\geq\epsilon)\leq\delta$. The constraint of $\epsilon$ comes from Theorem 3.1, and the elimination of $D^*$ in the constraint is by using $D^\*\geq0.5$ for any $n$ and $\phi$. It gives an theorectical upper bound on the number of utility evaluations required to achieve $P(\\|\hat\phi-\phi\\|_2^2\geq\epsilon)\leq\delta$, which is linear in $n$.
> - To emprically measure the scalability, we train decision trees on classification datasets so that $C=1$. As shown in Figure 1 that all curves are visibly parallel, and we emiprically confirm that the slope is $-0.5$ by fitting lines to them. This applies to other estimators included. It suggests that every curve can be fitted using $\log y=-\frac{1}{2}\log x+k$ where $y=\frac{1}{M}\sum_{m=1}^M\\|\hat\phi-\phi\\|_2$ ($M$ is the number of random seeds) and $x=\log T$. Then, $T\propto10^{2k}$ for achieving $y=\epsilon$ where $\epsilon$ is arbitrary. Therefore, we will use $10^{2k}$ as a time proxy, and compute the correlation coefficient ($r$) using 9 different values of $n$, i.e., $n \in \\{32,50,51,54,57,308,561,784,3072\\}$. Our experimental results do not present a significant increase in time proxy as $n$ increases, as shown below.
>
> Semi-value|0.1|0.2|0.3|0.4|0.5|0.6|0.7|0.8|0.9|(16,1)|(4,1)|(1,1)|(1,4)|(1,16)|(16,4)|(2,2)|(8,8)|(4,16)
> -|-|-|-|-|-|-|-|-|-|-|-|-|-|-|-|-|-|-
> $r$|-0.36|-0.32|-0.28|-0.24|-0.18|-0.11|-0.05|0.01|0.08|-0.35|-0.22|-0.07|0.01|0.11|-0.31|-0.11|-0.15|0.01
>
> Q3: A section highlighting the computational overhead of the various methods empirically would be helpful.
>
> Thanks for the suggestion. Assume we have sampled T pairs $\{(S, U(S))\}$, the overhead of different estimators are as follows:
> Estimator|Any linear-space estimator|least-square kernelSHAP|OFA
> -|:-:|:-:|:-:
> Overhead|$O(Tn)$|$O(Tn^2 + n^3)$|$Tn^2$

---

> > ### Author Rebuttal · Reviewer_AAnM · 2026-04-03
> >
> > I thank the authors for their detailed rebuttal. The authors have addressed my questions regarding scaling and have run additional experiments for the same. The rebuttal sufficiently clarifies my questions and I will maintain my current score.

---

### Official Review · Reviewer_8CR1 · 2026-03-13

**Soundness:** 4
**Presentation:** 4
**Significance:** 3
**Originality:** 3
**Overall Recommendation:** 5
**Confidence:** 3

**Summary:**

This paper studies efficient approximation of the Shapley value and more general semi-values under a linear-space constraint. The paper first uses a vector concentration viewpoint to sharpen asymptotic query complexity bounds for existing unbiased randomized estimators, removing an extra log n factor in the analysis of several existing unbiased randomized estimators and yields a unified framework that arises in individual estimations. It then introduces a unified framework in which semi-values are approximated by weighted random subset averages, derives the unique complexity-optimal subset-size distribution q* for minimizing the asymptotic query complexity. This framework characterizes when paired sampling helps, and shows how existing methods fit into this framework. Finally, the paper proposes Adalina, an adaptive linear-time linear-space estimator that centers utilities by an estimated baseline to reduce approximation variance while maintaining the query complexity.

**Compliance With Llm Reviewing Policy:**

Affirmed.

**Final Justification:**

I consider this to be a solid paper overall. It provides a clear discussion of the background, prior limitations, and the need for improvement, and the mathematical development is also presented in a reasonably understandable way for a theoretical paper. The rebuttal further helped by giving intuitive explanations for the parts I had found harder to follow, and I also found the discussion of the experimental choices reasonable. As a result, my understanding improved and I raised my evaluation accordingly.

**Key Questions For Authors:**

Overall, I found the paper technically solid and well motivated, and I believe a clear response on this point could further strengthen my assessment.

1. Since the proposed framework is designed to reduce asymptotic query complexity, could the authors provide empirical results to support this main claim such as the number of utility queries needed to reach a fixed error threshold? This would help me better assess how strongly the theoretical complexity advantage translates into practical gains.

2. Could the authors expand the intuitive explanation of the variance-reduction mechanism in Section 4? In particular, it would help to clarify why subtracting a scalar baseline reduces variance intuitively and how this mechanism is connected to the special structure induced by the optimal sampling distribution.

**Limitations:**

No. While this point is mentioned briefly in the paper, for example, that unbiased kernelSHAP can outperform Adalina in the standard Shapley-value setting, it would be better if such technical limitations were discussed more explicitly.

**Strengths And Weaknesses:**

Strengths

1. The paper provides a clear and well-structured overview of the problem setting, prior lines of work, and the remaining complexity bottlenecks in Shapley value estimation. Although the paper is theoretical, I found the overall motivation and problem setup reasonably readable, especially in how it positions linear-space randomized estimators and explains why sharper complexity analysis remains meaningful.

2. A major strength of the paper is the unified framework for approximating semi-values derived from a vector concentration inequality. The paper identifies the quantity D(q) that governs asymptotic query complexity, derives the corresponding optimal subset-size distribution q*. This framework also provides a useful perspective for connecting several existing methods and clarifying their optimality with respect to asymptotic query complexity.

3. It provides a theoretical interpretation of paired sampling, helping explain when this empirically used strategy is genuinely beneficial.

4. The Adalina algorithm combines the optimal sampling distribution with an adaptive centering strategy that reduces approximation variance while preserving the linear-space setting. I view this as a well-motivated and theoretically grounded refinement.

Weaknesses

1. Since the paper emphasizes asymptotic query complexity, it would have been useful to include experiments that directly measure the number of utility queries required to reach a fixed error threshold. Such results would make the claimed complexity improvement more directly visible in practice.

2. The key intuition behind the variance-reduction strategy in Section 4 is somewhat compressed. In particular, the paper could better explain why subtracting the scalar baseline gamma reduces variance intuitively, and why this mechanism depends on the special structure induced by the optimal sampling distribution.

---

> ### Author Rebuttal · Authors · 2026-03-31
>
> We thank Reviewer 8CR1 for the constructive feedback.
>
> Q1: Since the proposed framework is designed to reduce asymptotic query complexity, could the authors provide empirical results to support this main claim such as the number of utility queries needed to reach a fixed error threshold? This would help me better assess how strongly the theoretical complexity advantage translates into practical gains.
>
> - We have maken it explicitly in our revision that, for any $0<\epsilon\leq 2C$ (assume $\\|U\\|\_{\infty}\leq C$) and any $0<\delta<1$, Adalina requires $\frac{16nD^* C^2}{\epsilon^2}\log\frac{4}{\delta}$ utility evaluations to achieve $P(\\|\hat{\phi} - \\phi\\|_2^2\geq\epsilon)\leq\delta$. This theoretical gaurantee holds for every $n$. The constraint of $\epsilon$ comes from Theorem 3.1, and the elimination of $D^*$ in the constraint is by using $D^\* \geq0.5$ for any $n$ and semivalue.
> - Our experimental results show that there appears to have an approximation law that fits the performance of all the randomized algorithms included. This can be seen in Figure 1 that all curves are visibly parallel, and we emiprically confirm that the slope is -0.5 by fitting lines to them. It suggests that every curve can be fitted using $\log y=-\frac{1}{2}\log x+k$ where $y=\frac{1}{M}\sum_{m=1}^M \\|\hat{\phi} - \phi\\|_2$ ($M$ is the number of random seeds) and $x=\log T$. Then, given two estimators with $k_1$ and $k_2$, it implies that $\frac{T_1}{T_2}=10^{2(k_1-k_2)}$ where $T_i$ is the number of utility evaluations to achieve $y=\epsilon$ where $\epsilon$ is arbitrary. Based on this, we present $\frac{T\_{\mathrm{Adalina}}}{T\_{\mathrm{any}}}$ in the following, where we have included three additional datasets with larger $n$.
>
> **philippine, n=308**||||||||||
> -|-|-|-|-|-|-|-|-|-
> **Estimator**|**(16,1)**|**(4,1)**|**(1,1)**|**(1,4)**|**(1,16)**|**(16,4)**|**(2,2)**|**(8,8)**|**(4,16)**
> MSR-Witter|0.99|0.95|0.61|0.21|0.06|0.96|0.65|0.68|0.27
> SHAP-IQ|0.40|0.47|0.50|0.09|0.02|0.16|0.29|0.11|0.05
> kernelSHAP|||1.80||||||
> AME|0.96|1.08|0.21|0.05|0.02|0.94|0.54|0.68|0.24
> ARM|0.22|0.25|0.52|0.29|0.25|0.50|0.50|0.58|0.45
> GELS|0.24|0.28|0.23|0.26|0.22|0.44|0.36|0.58|0.53
> GELS-Shapley|||0.38||||||
> **Fashion-MNIST, n=783**
> MSR-Witter|0.05|0.17|0.54|0.90|0.98|0.25|0.56|0.54|0.88
> SHAP-IQ|0.35|0.37|0.31|0.04|0.00|0.13|0.15|0.06|0.02
> kernelSHAP|||0.25||||||
> AME|0.02|0.06|0.26|1.40|1.62|0.22|0.52|0.53|0.95
> ARM|0.21|0.25|0.46|0.20|0.15|0.50|0.68|0.69|0.49
> GELS|0.16|0.28|0.33|0.16|0.20|0.46|0.35|0.68|0.51
> GELS-Shapley|||0.44||||||
> **CIFAR_10_small, n=3,072**
> MSR-Witter|0.01|0.09|0.50|0.79|0.93|0.06|0.40|0.29|0.66
> SHAP-IQ|0.34|0.47|0.52|0.31|0.23|0.10|0.26|0.10|0.12
> kernelSHAP|||0.32||||||
> AME|0.00|0.04|0.36|0.92|1.19|0.06|0.39|0.29|0.66
> ARM|0.17|0.18|0.34|0.24|0.19|0.51|0.50|0.38|0.40
> GELS|0.20|0.21|0.22|0.19|0.16|0.44|0.61|0.61|0.64
> GELS-Shapley|||0.36||||||
> **har, n=561**
> MSR-Witter|0.08|0.29|0.35|0.12|0.04|0.23|0.33|0.28|0.17
> SHAP-IQ|0.44|0.52|0.43|0.08|0.02|0.13|0.21|0.07|0.04
> kernelSHAP|||2.66||||||
> AME|0.03|0.20|0.13|0.04|0.01|0.24|0.28|0.28|0.15
> ARM|0.18|0.17|0.30|0.28|0.23|0.51|0.68|0.66|0.57
> GELS|0.33|0.25|0.17|0.20|0.30|0.48|0.42|0.45|0.67
> GELS-Shapley|||0.35||||||
>
> Q2: intuitive explanation of the variance-reduction mechanism in Section 4
> - We note first that $\phi(U-\gamma1)=\phi(U)$ for any $\phi$, $\gamma$ and $U$, as $\phi(\gamma1)\equiv0$. Here, $1$ represents the all-one utility.
> - The estimator $\hat\phi^\gamma$ in Section 4 can be decomposed into two terms $\bar X+\gamma\bar Y$, where $\bar X$ is an unbiased estimate of $\phi(U)$, the objective of interest, while $\bar Y$ is an unbiased estimate of 0. The key is that $\bar X$ and $\bar Y$ are **correlated** so that by choosing $\gamma$ approporately we can minimize the mean square error
> $$
> E\\|\bar X+\gamma\bar Y-\phi(U)\\|_2^2 = E\\|\bar X-\phi(U)\\|_2^2+\gamma^2 E\\|\bar Y\\|_2^2+2\gamma E[(\bar X-\phi(U)\cdot\bar Y].
> $$
> Had $\bar X$ and $\bar Y$ been independent, the last term would vanish and the optimal choice for $\gamma$ would be 0. However, since we choose *correlated* $\bar X$ and $\bar Y$, such that the last expectation is nonzero, making the optimal $\gamma$ nonzero.
> - With the optimal sampling distribution $q^\*$, we have $E[\\|u_S z_S-\gamma z_S - \varphi\\|_{2}^{2}] = nD^\* E[(u_S-\gamma)^2]-\\|\varphi\\|_2^2$, as follows from Theorem 3.1. This identity shows that the mean square error $E\\|\bar X+\gamma\bar Y-\phi(U)\\|_2^2$ is minimized precisely when $\gamma^\*=E[u_S]$. Since we used the same sampling distribution $q^\*$ to derive $\bar X$ and $\bar Y$, we can re-use the sampled sets $S_t$ to estimate $\gamma^\*=E[u_S]$, which leads to our algorithm Adalina. Note that the resulting estimator becomes biased but its mean square error, as shown in Theorem 4.1, matches the oracle bound when $\gamma^\*$ is known in advance, up to an higher order term.

---

> > ### Author Rebuttal · Reviewer_8CR1 · 2026-04-03
> >
> > Thank you for the clear and intuitive rebuttal. It helped me better understand the intuition behind the variance-reduction mechanism. The explanation regarding the experimental results on asymptotic query complexity was also reasonable. Based on these clarifications, I have raised my score accordingly.

---

### Official Review · Reviewer_YShk · 2026-04-13

**Soundness:** 3
**Presentation:** 3
**Significance:** 3
**Originality:** 3
**Overall Recommendation:** 4
**Confidence:** 4

**Summary:**

The article addresses the problem of approximating semi-values, establishing improved theoretical results on the asymptotic number of queries required to achieve $(\varepsilon,\delta)$- approximations while minimizing the approximation variance. The main contribution relies on a vector concentration inequality, also developed by the authors.

The article applies this theoretical framework to several SOTA methods, obtaining sharper bounds on their asymptotic query complexity. It then proposes an adaptive algorithm for approximating semi-values, based on the insights derived from the theoretical analysis. Finally, the authors conduct an extensive set of experiments to empirically compare the proposed algorithm against several SOTA methods, showing strong performance across all considered scenarios.

**Compliance With Llm Reviewing Policy:**

Affirmed.

**Final Justification:**

The article is well written and presents solid contributions. Accordingly, I recommend its acceptance. That said, I believe the authors could have done a better job of positioning their work within the broad range of machine learning tasks to which semi-values can be applied. For this reason, I recommend a weak accept.

**Key Questions For Authors:**

I do not have questions.

**Limitations:**

The article does not discuss its limitations. Semi-values are applied to different settings in machine learning, including feature-importance, data valuation, and dataset valuation. The article never places itself in one of them, and the nuances of the different settings are not properly addressed, which can produce confusion to the reader.

**Strengths And Weaknesses:**

**Strengths.**
 - The article is well written and clearly structured.
 - The article positions itself in the context of prior and concurrent literature and clearly discuss how it differs.
 - The article is rigorous without being unnecessarily technical.
 - The considered problem is of great interest and the article is a clear fit for ICML.
 - The theoretical results of the article are sound and non trivial.
 - The article presents a rich collection of experiments that support the theoretical results.
 - The vector concentration inequality presented in the article is a great contribution which might help future research.

**Weaknesses.**

 - The article does not discuss the limitations of its model or results. For instance, it does not address the complexity of computing the sampling distribution in Algorithm 1, nor does it clearly and explicitly state constraints such as $\varepsilon \leq \frac{3\sigma^2}{C}$ in Theorem 2.1.
 - The article is sometimes unclear about the precise setting in which it operates, namely, whether it relates to feature importance, data valuation, dataset valuation, etc. While this may be intentional, in order to emphasize the generality of the results, the combination of methods across different application domains can be confusing for the reader. This is particularly the case given the growing variety of semi-value-based methods in the machine learning literature, which makes it difficult to keep track of all variants without explicit clarification. Moreover, the empirical results are only on feature importance, which makes wonder about the performances of the proposed method in other tasks.

---

### Decision · Program_Chairs · 2026-04-30

**Decision:**

Accept (regular)

**Comment:**

The paper looks at approximation of Shapley value (and other semi-values) with the constraint of having linear space complexity, looking at both variance reduction and query complexity. There are two main contributions:
- the paper establishes a framework based on a vector concentration inequality (that avoids using union bounds) that enables proving tighter bounds for several existing approximation methods. This result sheds light on performance of multiple previous algorithms.
- the paper proposes Adalina, an adaptive algorithm that achieves theoretically proven variance reduction for any utility function.
The paper also contains empirical validations of the proposed results.

All the reviewers found the results timely and interesting, in particular for the light they shed on a topic on which many papers have been written in the past few years. Despite this, Reviewer 81zp remained negative, for the following main reasons:
1. the theoretical framework (based on the vector concentration inequality) is straightforward;
2. the empirical validation had an issue in the comparison with Witter et al 2025;
3. the reviewer proposed an alternative method based on sampling without replacement that could potentially achieve a similar variance reduction.

The author-reviewer discussion period did not allow to settle these points. I looked at all three myself and found that
1. it is true that the vector concentration inequality is a simple derivative of a result that can be found in (Yurinsky, 1995)---the authors do not pretend otherwise. However, using this vector concentration inequality is a new perspective that was not used in the semi-value estimation literature from the past years, and it does bring better results than what was known.[*] So in my opinion it is a perspective worth publishing even though it is not technically difficult.
2. it is true that the initial paper was not correctly implementing the technique from (Witter et al, 2025). This has been fixed during the rebuttal and it appears that (Witter et al, 2025) works better now. As the authors argue, there are good reasons for this, namely that they used a tree-based utility. Also, the comparison is not strictly fair given that (Witter et al, 2025) is not linear space. All-in-all, my take on this is that it is not really key whether Adalina is better or worse than (Witter et al, 2025); but it is natural to expect a more in-depth comparison of the two in any revision.
3. regarding sampling w/o replacement, the reviewer's proposal is interesting and I encourage the authors to think about it further. However, there are some difficulties to overcome (in particular due to non-uniform distributions used in Adalina) and it is not clear whether it would work and what the results would be. In any case, the fact that there is potentially another method to achieve something similar (to be seen) is scientifically interesting but cannot be held against the current submission.

In summary, the paper provides interesting results on a timely topic (even though they are not technically difficult), which unifies previous contributions. It is also well written. So it is a valid contribution to ICML and can be accepted as such. On the negative side, given the nature of the contribution, one would appreciate a more detailed experimental comparison to (Witter et al 2025).


[*] This includes a theoretical understanding of paired sampling, which I find interesting. This finding is contemporary to https://arxiv.org/pdf/2602.01399 (appeared on arxiv post-deadline) and in fact is also complementary.